# Dual Cognitive Architecture: Incorporating Biases and Multi-Memory Systems for Lifelong Learning

**Shruthi Gowda**[1,2]**, Bahram Zonooz**[†2,3]**, Elahe Arani**[†2]
*s.gowda@tue.nl, b.zonooz@tue.nl, e.arani@gmail.com*
[1]*NavInfo Europe*     [2]*Eindhoven University of Technology*     [3]*TomTom*
[†]*Contributed equally.*

**Reviewed on OpenReview:** *https://openreview.net/forum?id=PEyVq0hlO3*

## Abstract

Artificial neural networks (ANNs) exhibit a narrow scope of expertise on stationary independent data. However, the data in the real world is continuous and dynamic, and ANNs must adapt to novel scenarios while also retaining the learned knowledge to become lifelong learners. The ability of humans to excel at these tasks can be attributed to multiple factors ranging from cognitive computational structures, cognitive biases, and the multi-memory systems in the brain. We incorporate key concepts from each of these to design a novel framework, *Dual Cognitive Architecture (DUCA)*, which includes multiple sub-systems, implicit and explicit knowledge representation dichotomy, inductive bias, and a multi-memory system. The inductive bias learner within DUCA is instrumental in encoding shape information, effectively countering the tendency of ANNs to learn local textures. Simultaneously, the inclusion of a semantic memory submodule facilitates the gradual consolidation of knowledge, replicating the dynamics observed in fast and slow learning systems, reminiscent of the principles underpinning the complementary learning system in human cognition. DUCA shows improvement across different settings and datasets, and it also exhibits reduced task recency bias, without the need for extra information. To further test the versatility of lifelong learning methods on a challenging distribution shift, we introduce a novel domain-incremental dataset *DN4IL*. In addition to improving performance on existing benchmarks, DUCA also demonstrates superior performance on this complex dataset. [1] [2]

## 1 Introduction

Deep learning has seen rapid progress in recent years, and supervised learning agents have achieved superior performance on perception tasks. However, unlike a supervised setting, where data is static and independent and identically distributed, real-world data is changing dynamically. Continual learning (CL) aims to learn multiple tasks when data is streamed sequentially (Parisi et al., 2019). This is crucial in real-world deployment settings, as the model needs to adapt quickly to novel data (plasticity), while also retaining previously learned knowledge (stability). Artificial neural networks (ANN), however, are still not effective lifelong learners, as they often fail to generalize to small changes in distribution and also suffer from forgetting old information when presented with new data (catastrophic forgetting) (McCloskey & Cohen, 1989).

Humans, on the other hand, show a better ability to acquire new skills while also retaining previously learned skills to a greater extent. This intelligence can be attributed to different factors in human cognition. Multiple theories have been proposed to formulate an overall cognitive architecture, which is a broad domain-generic cognitive computation model that captures the essential structure and process of the mind. Some of these theories hypothesize that, instead of a single standalone module, multiple modules in the brain share information to excel at a particular task. CLARION (Connectionist learning with rule induction online)

---

[1]Code is public at `https://github.com/NeurAI-Lab/DUCA`.
[2]*DN4IL* dataset is public at `https://github.com/NeurAI-Lab/DN4IL-dataset`.

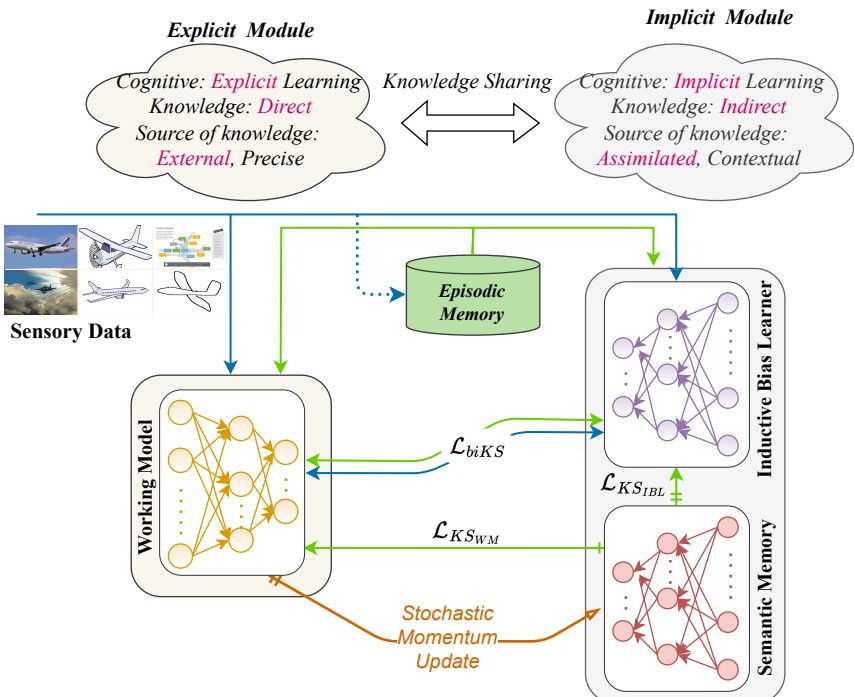

Figure 1: Schematic of *Dual Cognitive Architecture (DUCA)*. The working model in the explicit module learns direct sensory data. Within the implicit module, the inductive bias learner encodes the prior shape knowledge and the semantic memory consolidates information from the explicit module. Only one network (semantic memory) is used during inference as it includes consolidated knowledge across all tasks.

(Sun & Franklin, 2007) is one such theory that postulates an integrative cognitive architecture, consisting of a number of distinct subsystems. It predicates a dual representational structure (Chaiken & Trope, 1999), where the top level encodes conscious explicit knowledge, while the other encodes indirect implicit information. The two systems interact, share knowledge, and cooperate to solve tasks. Delving into these underlying architectures and formulating a new design can help in the quest to build intelligent agents.

Multiple modules can be instituted instead of a single feedforward network: an explicit module responsible for learning from the standard visual input and an implicit module that specializes in acquiring and sharing contextual knowledge indirectly. The implicit module can be further divided into more submodules, each providing different information. Inductive biases and semantic memories can act as different kinds of implicit knowledge. Inductive biases are pre-stored templates or knowledge that provide some meaningful disposition toward adapting to the continuously evolving world (Chollet, 2019). Theories postulate that after rapidly learning information, a gradual consolidation of knowledge transpires in the brain for slow learning of structured information (Kumaran et al., 2016). Thus, the new design incorporates multiple concepts of cognition architectures, the dichotomy of implicit and explicit representations, inductive biases, and multi-memory systems theory.

To this end, we propose *Dual Cognitive Architecture* (DUCA), a multi-module architecture for CL. The explicit working module processes the standard input data. Two different submodules are introduced for the implicit module. The inductive bias learner embeds relevant prior information, and as networks are shown to be biased toward textural information (unlike humans that are more biased toward global semantics) (Geirhos et al., 2019), we propose to utilize global shape information as the prior. Both texture and shape are present in the original image, but ANNs tend to rely more on texture and ignore semantic information. Hence, we utilize the implicit shape information and share it with the explicit module to learn more generic and high-level representations. Further, to emulate the consolidation of information in the slow-fast multi-memory system, a gradual accumulation of knowledge from the explicit working module is embedded in the

second semantic memory submodule. We show that interacting and leveraging information between these modules can help alleviate catastrophic forgetting, while also increasing the robustness to the distribution shift.

DUCA achieves superior performance across all CL settings on various datasets. DUCA outperforms SOTA CL methods on Seq-CIFAR10 and Seq-CIFAR100 in class-incremental settings. Furthermore, in more realistic general class-incremental settings where the task boundary is blurry and the classes are not disjoint, DUCA shows significant gains. The addition of inductive bias and semantic memory helps to achieve a better balance between the plasticity-stability trade-off. The prior in the form of shape helps produce generic representations, and this results in DUCA exhibiting a reduced task-recency bias. Furthermore, DUCA also shows greater robustness against natural corruption. Finally, to test the capability of CL methods against the distribution shift, we introduce a domain-incremental learning dataset, *DN4IL*, which is a carefully designed subset of the DomainNet dataset (Peng et al., 2019). DUCA shows considerable robustness across all domains on these challenging data, thus establishing the efficacy of our cognitive-inspired CL architecture. Our contributions are as follows:

- *Dual Cognitive Architecture (DUCA)*, a novel method that incorporates aspects of cognitive architectures, multi-memory systems, and inductive bias into the CL framework.
- Introducing *DN4IL*, a challenging domain-incremental learning dataset.
- Benchmark across different CL settings: class-, task-, generalized class-, and domain-incremental learning.
- Analyses on the plasticity-stability trade-off, task recency bias, and robustness to natural corruptions.

## 2 Methodology

### 2.1 Cognitive Architectures

Cognitive architectures refer to computational models that encapsulate the overall structure of the cognitive process in the brain. The underlying infrastructure of such a model can be leveraged to develop better intelligent systems. Global workspace theory (GWT) (Juliani et al., 2022) postulates that human cognition is composed of a multitude of special-purpose processors and is not a single standalone module. Different sub-modules might encode different contextual information which, when activated, can transfer knowledge to the conscious central workspace to influence and help make better decisions. Furthermore, CLARION (Sun & Franklin, 2007) posits a dual-system cognitive architecture with two levels of knowledge representation. The explicit module encodes direct knowledge that is externally accessible. The implicit module encodes indirect knowledge that is not directly accessible, but can be obtained through some intermediate interpretive or transformational steps. These two modules interact with each other by transferring knowledge between each other.

Inspired by these theories, we formulate a method that incorporates some of the key aspects of cognitive architecture into the CL method. A working module, which encodes the direct sensory data, forms the explicit module. A second module that encodes indirect and interpretive information forms the implicit module. The implicit module further includes multiple sub-modules to encode different types of knowledge.

### 2.2 Inductive Bias

The sub-modules in the implicit module need to encapsulate implicit information that can provide more contextual and high-level supervision. One of such knowledge can be prior knowledge or inductive bias. Inductive biases are pre-stored templates that exist implicitly even in earlier stages of the human brain (Pearl & Mackenzie, 2018). For instance, cognitive inductive bias may be one of the reasons why humans can focus on the global semantics of objects to make predictions. ANNs, on the other hand, are more prone to rely on local cues and textures (Geirhos et al., 2019). Global semantics or shape information already exists in the visual data, but in an indirect way. The incorporation of shape-awareness to the networks has proven to be a more effective approach in acquiring generic representations (Gowda et al., 2022). Hence, we

utilize shape as indirect information in the implicit module. The sub-module uses a transformation step to extract the shape and share this inductive bias with the working module. As the standard (RGB) image and its shape counterpart can be viewed as different perspectives/modalities of the same data, ensuring that the representation of one modality is consistent with the other increases robustness to spurious correlations that might exist in only one of them.

## 2.3   Multi Memory System

Moreover, many theories have postulated that an intelligent agent must possess differentially specialized learning memory systems (Kumaran et al., 2016). While one system rapidly learns the individual experience, the other gradually assimilates the knowledge. To emulate this behavior, we establish a second sub-module that slowly consolidates the knowledge from the working module.

## 2.4   Formulation

To this end, we propose a novel method *Dual Cognitive Architecture (DUCA)*, which incorporates all these concepts into the CL paradigm. DUCA consists of two modules, the explicit module, and the implicit module. The explicit module has a single working model and processes the incoming direct visual data. The implicit module further consists of two submodules, namely the inductive bias learner and the semantic memory. They share relevant contextual information and assimilated knowledge with the explicit module, respectively. Figure 1 shows the overall architecture.

In the implicit module, semantic memory $N_{SM}$, consolidates knowledge at stochastic intervals from the working model $N_{WM}$, in the explicit module. The other submodule, the inductive bias learner $N_{IBL}$, processes the data and extracts shape information (Section G). $N_{WM}$ processes the RGB data, $N_{SM}$ consolidates the information from the working module at an update frequency in a stochastic manner, and $N_{IBL}$ learns from the shape data. The encoder or the feature extractor network takes an image as the input and produces latent representations, which are then passed to the linear classifier to do object recognition. $f$ represents the combination of the encoder and the classifier, and $\theta_{WM}$, $\theta_{SM}$, and $\theta_{IBL}$ are the parameters of the three networks.

A CL classification consists of a sequence of $T$ tasks and, during each task, $t \in 1, 2...T$, samples $x_c$ and their corresponding labels $y_c$ are drawn from the current task data $D_t$. Furthermore, for each subsequent task, a random batch of exemplars is sampled from episodic memory $B$ as $x_b$. An inductive bias (shape) filter is applied to generate shape samples, $x_{c_s} = \mathbb{IB}(x_c)$ and $x_{b_s} = \mathbb{IB}(x_b)$. Reservoir sampling (Vitter, 1985) is incorporated to replay previous samples. Each of the networks $N_{WM}$ and $N_{IBL}$ learns in its own modality with a supervised cross-entropy loss on both the current samples and the buffer samples:

$$
\begin{aligned}
\mathcal{L}_{Sup_{WM}} &= \mathcal{L}_{CE}(f(x_c; \theta_{WM}), y_c) + \mathcal{L}_{CE}(f(x_b; \theta_{WM}), y_b) \\
\mathcal{L}_{Sup_{IBL}} &= \mathcal{L}_{CE}(f(x_{c_s}; \theta_{IBL}), y_c) + \mathcal{L}_{CE}(f(x_{b_s}; \theta_{IBL}), y_b)
\end{aligned}
\tag{1}
$$

The Knowledge Sharing (KS) objectives are designed to transfer and share information between all modules. KS occurs for current samples and buffered samples. We employ the mean squared error as the objective function for all KS losses. To provide shape supervision to the working model and vice versa, a bidirectional decision space similarity constraint ($\mathcal{L}_{biKS}$) is enforced to align the features of the two modules.

$$
\mathcal{L}_{biKS} = \underset{x \sim D_t \cup B}{\mathbb{E}} \| f(x_s; \theta_{IBL}) - f(x; \theta_{WM}) \|_2^2
\tag{2}
$$

The consolidated structural information in semantic memory is transferred to both the working model and the inductive bias learner by aligning the output space on the buffer samples, which further helps in information retention. The loss functions $\mathcal{L}_{KS_{WM}}$ and $\mathcal{L}_{KS_{IBL}}$ are as follows;

$$
\begin{aligned}
\mathcal{L}_{KS_{WM}} &= \underset{x_b \sim B}{\mathbb{E}} \| f(x_b; \theta_{SM}) - f(x_b; \theta_{WM}) \|_2^2 \\
\mathcal{L}_{KS_{IBL}} &= \underset{x_b \sim B}{\mathbb{E}} \| f(x_b; \theta_{SM}) - f(x_{b_s}; \theta_{IBL}) \|_2^2
\end{aligned}
\tag{3}
$$

Table 1: Comparison of different methods on standard CL benchmarks (Class-IL, Task-IL and GCIL settings). DUCA shows a consistent improvement over all methods for both buffer sizes.

| $|\mathcal{B}|$ | METHOD | SEQ-CIFAR10 | | SEQ-CIFAR100 | | GCIL-CIFAR100 | |
|---|---|---|---|---|---|---|---|
| | | CLASS-IL | TASK-IL | CLASS-IL | TASK-IL | UNIFORM | LONGTAIL |
| - | JOINT | $92.20_{\pm0.15}$ | $98.31_{\pm0.12}$ | $70.62_{\pm0.64}$ | $86.19_{\pm0.43}$ | $60.45_{\pm1.65}$ | $60.10_{\pm0.42}$ |
| | SGD | $19.62_{\pm0.05}$ | $61.02_{\pm3.33}$ | $17.58_{\pm0.04}$ | $40.46_{\pm0.99}$ | $10.36_{\pm0.13}$ | $9.62_{\pm0.21}$ |
| 200 | ER | $44.79_{\pm1.86}$ | $91.19_{\pm0.94}$ | $21.40_{\pm0.22}$ | $61.36_{\pm0.39}$ | $16.52_{\pm0.10}$ | $16.20_{\pm0.30}$ |
| | DER++ | $64.88_{\pm1.17}$ | $91.92_{\pm0.60}$ | $29.60_{\pm1.14}$ | $62.49_{\pm0.78}$ | $27.73_{\pm0.93}$ | $26.48_{\pm2.04}$ |
| | Co$^2$L | $65.57_{\pm1.37}$ | $93.43_{\pm0.78}$ | $31.90_{\pm0.38}$ | $55.02_{\pm0.36}$ | - | - |
| | ER-ACE | $62.08_{\pm1.44}$ | $92.20_{\pm0.57}$ | $32.49_{\pm0.95}$ | $59.77_{\pm0.31}$ | $27.64_{\pm0.76}$ | $25.10_{\pm2.64}$ |
| | CLS-ER | $66.19_{\pm0.75}$ | $93.90_{\pm0.60}$ | $43.80_{\pm1.89}$ | $73.49_{\pm1.04}$ | $35.88_{\pm0.41}$ | $35.67_{\pm0.72}$ |
| | DUCA | $\mathbf{70.04_{\pm1.07}}$ | $\mathbf{94.49_{\pm0.38}}$ | $\mathbf{45.38_{\pm1.28}}$ | $\mathbf{76.62_{\pm0.16}}$ | $\mathbf{38.61_{\pm0.83}}$ | $\mathbf{37.11_{\pm0.16}}$ |
| 500 | ER | $57.74_{\pm0.27}$ | $93.61_{\pm0.27}$ | $28.02_{\pm0.31}$ | $68.23_{\pm0.16}$ | $23.62_{\pm0.66}$ | $22.36_{\pm1.27}$ |
| | DER++ | $72.70_{\pm1.36}$ | $93.88_{\pm0.50}$ | $41.40_{\pm0.96}$ | $70.61_{\pm0.11}$ | $35.80_{\pm0.62}$ | $34.23_{\pm1.19}$ |
| | Co$^2$L | $74.26_{\pm0.77}$ | $95.90_{\pm0.26}$ | $39.21_{\pm0.39}$ | $62.98_{\pm0.58}$ | - | - |
| | ER-ACE | $68.45_{\pm1.78}$ | $93.47_{\pm1.00}$ | $40.67_{\pm0.06}$ | $66.45_{\pm0.71}$ | $30.14_{\pm1.11}$ | $31.88_{\pm0.73}$ |
| | CLS-ER | $75.22_{\pm0.71}$ | $94.94_{\pm0.53}$ | $51.40_{\pm1.00}$ | $78.12_{\pm0.24}$ | $38.94_{\pm0.38}$ | $38.79_{\pm0.67}$ |
| | DUCA | $\mathbf{76.20_{\pm0.70}}$ | $\mathbf{95.95_{\pm0.14}}$ | $\mathbf{54.27_{\pm1.09}}$ | $\mathbf{79.80_{\pm0.32}}$ | $\mathbf{43.34_{\pm0.32}}$ | $\mathbf{41.44_{\pm0.22}}$ |

Thus, the overall loss functions for the working model and the inductive bias learner are as follows;

$$
\begin{aligned}
\mathcal{L}_{WM} &= \mathcal{L}_{Sup_{WM}} + \lambda(\mathcal{L}_{biKS} + \mathcal{L}_{KS_{WM}}) \\
\mathcal{L}_{IBL} &= \mathcal{L}_{Sup_{IBL}} + \gamma(\mathcal{L}_{biKS} + \mathcal{L}_{KS_{IBL}})
\end{aligned}
\tag{4}
$$

The semantic memory of the implicit module is updated with a stochastic momentum update (SMU) of the weights of the working model at rate $r$ with a decay factor of $d$,

$$
\theta_{SM} = d \cdot \theta_{SM} + (1-d) \cdot \theta_{WM} \text{ if } s \sim U(0,1) < r
\tag{5}
$$

More details are provided in Algorithm 1. We discuss the computational aspect in Section F. Note that we use semantic memory ($\theta_{SM}$) for inference, as it contains consolidated knowledge across all tasks.

## 3 Experimental Settings

ResNet-18 (He et al., 2016) architecture is used for all experiments. All networks are trained using the SGD optimizer with standard augmentations of random crop and random horizontal flip. The different hyperparameters, tuned per dataset, are provided in E. The different CL settings are explained in detail in Section D. We consider CLass-IL, Domain-IL, and also report the Task-IL settings. Seq-CIFAR10 and Seq-CIFAR100 (Krizhevsky et al., 2009) for the class-incremental learning (Class-IL) settings, which are divided into 5 tasks each. In addition to Class-IL, we also consider and evaluate general Class-IL (GCIL) (Mi et al., 2020) on the CIFAR100 dataset. For domain-incremental learning (Domain-IL), we propose a novel dataset, *DN4IL*.

## 4 Results

We provide a comparison of our method with standard baselines and multiple other SOTA CL methods. The lower and upper bounds are reported as SGD (standard training) and JOINT (training all tasks together), respectively. We compare with other rehearsal-based methods in the literature, namely ER, DER++ (Buzzega et al., 2020), Co$^2$L (Cha et al., 2021), ER-ACE (Caccia et al., 2021), and CLS-ER (Arani et al., 2022). Table 1 shows the average performance in different settings over three seeds. Co$^2$L utilizes task boundary information, and therefore the GCIL setting is not applicable. The results are taken from the

original works and, if not available, using the original codes, we conducted a hyperparameter search for the new settings (see Section E for details).

DUCA achieves the best performance across all datasets in all settings. In the challenging Class-IL setting, we observe a gain of ∼50% over DER++, thus showing the efficacy of adding multiple modules for CL. Furthermore, we report improvements of ∼6% on both the Seq-CIFAR10 and Seq-CIFAR100 datasets, over CLS-ER, which utilizes two semantic memories in its design. DUCA has a single semantic memory, and the additional boost is obtained by prior knowledge from the inductive bias learner. Improvement is prominent even when the memory budget is low (200 buffer size). GCIL represents a more realistic setting, as the task boundaries are blurry, and classes can reappear and overlap in any task. GCIL-Longtail version also introduces an imbalance in the sample distribution. DUCA shows a significant improvement on both versions of GCIL-CIFAR100. Additional results are provided in Table 4.

Shape information from the inductive bias learner offers the global high-level context, which helps in producing generic representations that are not biased towards learning only the current task at hand. Furthermore, sharing of the knowledge that has been assimilated through the appearance of overlapping classes through the training scheme, further facilities learning in this general setting. The overall results indicate that the dual knowledge sharing between the explicit working module and the implicit inductive bias and semantic memory modules enables both better adaptation to new tasks and information retention.

## 5 Domain-incremental learning

Intelligent agents deployed in real-world applications need to maintain consistent performance through changes in the data and environment. Domain-IL aims to assess the robustness of the CL methods to the distribution shift. In Domain-IL, the classes in each task remain the same, but the input distribution changes, and this makes for a more plausible use case for evaluation. However, the datasets used in the literature do not fully reflect this setting. For instance, the most common datasets used in the literature are different variations (Rotated and Permuted) of the MNIST dataset (LeCun et al., 1998). MNIST is a simple dataset, usually evaluated on MLP networks, and its variations do not reflect the real-world distribution shift challenges that a CL method faces (the results for R-MNIST are presented in the Table C). Farquhar & Gal (2018) propose fundamental desiderata for CL evaluations and datasets based on real-world use cases. One of the criteria is to possess cross-task resemblances, which Permuted-MNIST clearly violates. Thus, a different dataset is needed to test the overall capability of a CL method to handle the distributional shift.

### 5.1 *DN4IL* Dataset

To this end, we propose *DN4IL* (DomainNet for Domain-IL), which is a well-crafted subset of the standard DomainNet dataset (Peng et al., 2019), used in domain adaptation. DomainNet consists of common objects in six different domains - real, clipart, infograph, painting, quickdraw, and sketch. The original DomainNet consists of 59k samples with 345 classes in each domain. The classes have redundancy, and moreover, evaluating the whole dataset can be computationally expensive in a CL setting. *DN4IL* version considers different criteria such as relevance of classes, uniform sample distribution, computational complexity, and ease of benchmarking for CL.

All classes were grouped into semantically similar supercategories. Of these, a subset of classes was selected that had relevance to domain shift, while also having maximum overlap with other standard datasets such as CIFAR, to facilitate out-of-distribution analyses. 20 supercategories were chosen with 5 classes each (resulting in a total of 100 classes). In addition, to provide a balanced dataset, we performed a class-wise sampling. First, we sample images per class in each supercategory and maintain class balance. Second, we choose samples per domain, so that it results in a dataset that has a near-uniform distribution across all classes and domains. The final dataset *DN4IL* is succinct, more balanced, and more computationally efficient for benchmarking, thus facilitating research in CL. Furthermore, the new dataset is deemed more plausible for real-world settings and also adheres to all evaluation desiderata by (Farquhar & Gal, 2018). The challenging distribution shift between domains provides an apt dataset to test the capability of CL methods

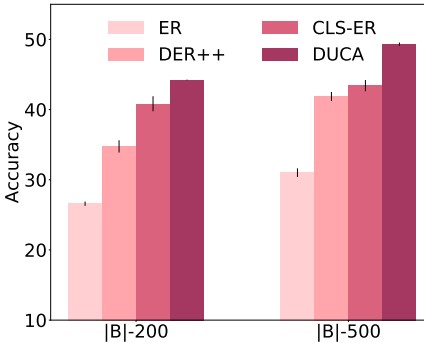 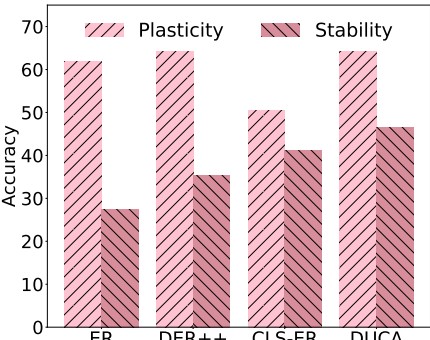

Figure 2: Accuracy (left) and plasticity-stability analysis (right) on *DN4IL* dataset. DUCA substantially outperforms other methods and with better plasticity-stability trade-off.

in the Domain-IL setting. More details, statistics, and visual examples of this crafted dataset are provided in Section H.

## 5.2 DN4IL Performance

Figure 2 (left) reports the results on *DN4IL* for two different buffer sizes (values are provided in Table 10). DUCA shows a considerable performance gain in the average accuracy across all domains, and this can be primarily attributed to the supervision from the shape data. Standard networks tend to exhibit texture bias and learn background or spurious cues (Geirhos et al., 2019) that result in performance degradation when the distribution changes. Learning global shape information of objects, on the other hand, helps in learning generic features that can translate well to other distributions. Semantic memory further helps to consolidate information across domains. Maintaining consistent performance to such difficult distribution shifts prove beneficial in real-world applications, and the proficiency of DUCA in this setting can thus open up new avenues for research in cognition-inspired multi-module architectures.

# 6 Model Analyses

## 6.1 Plasticity-Stability Trade-off

Plasticity refers to the capability of a model to learn new tasks, while stability shows how well it can retain old information. The plasticity-stability dilemma is a long-standing problem in CL, which requires an optimal balance between the two. Following Sarfraz et al. (2022), we measure each of these to assess the competence of the CL methods. Plasticity is computed as the average performance of each task when first learned (e.g., the accuracy of the network trained on task $T_2$, evaluated on the test set of $T_2$). Stability is computed as the average performance of all tasks $1 : T\text{-}1$, after learning the final task $T$. Figure 2 (right) reports these numbers for the *DN4IL* dataset. As seen, the ER and DER methods exhibit forgetting and lower stability, and focus only on the newer tasks. CLS-ER shows greater stability, but at the cost of reduced plasticity. However, DUCA shows the highest stability while maintaining comparable plasticity. The shape knowledge helps in learning generic solutions that can translate to new tasks, while the semantic consolidation update at stochastic rates acts as a regularization to maintain stable parameter updates. Thus, DUCA strikes a better balance between plasticity and stability.

## 6.2 Recency-Bias Analysis

Recency bias is a behavior in which the model predictions tend to be biased toward the current or the most recent task (Wu et al., 2019). This is undesirable in a CL model, as it results in a biased solution that forgets the old tasks. To this end, after the end of the training, we evaluate the models on the test set (of

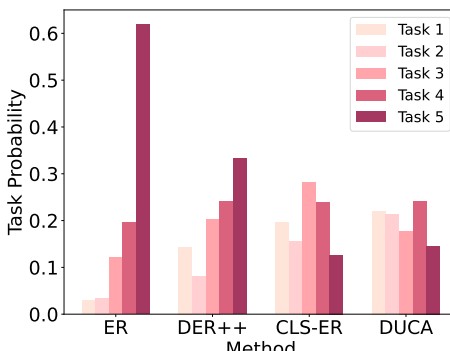 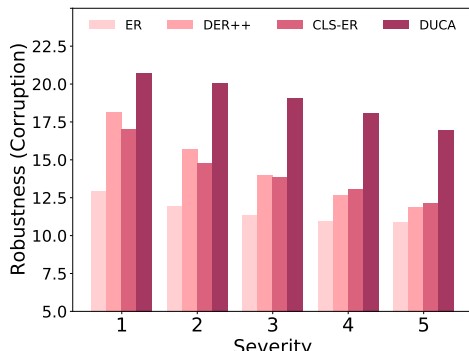

Figure 3: DUCA shows reduced task recency bias (left), as well as higher robustness against natural corruption (right) on Seq-CIFAR10 ($|\mathcal{B}|$=200) dataset.

all tasks) and calculate the probability of predicting each task. The output distribution for each test sample is calculated for all classes and the probabilities are averaged per task.

Figure 3 (left) shows the probabilities for each task on the Seq-CIFAR10 dataset. As shown, the ER and DER++ methods tend to incline most of their predictions toward the classes seen in the last task, thus creating a misguided bias. DUCA shows a lower bias compared to both of these baselines. CLS-ER exhibits reduced bias due to the presence of multiple memories, but the distribution is still relatively skewed (with respect to a probability of 0.2). DUCA shows a more uniform distribution across all tasks. The dual information from the shape data and the consolidated knowledge across tasks helps in breaking away from Occam's razor pattern of neural networks to default to the easiest solution.

## 6.3 Robustness

Lifelong agents, when deployed in real-world settings, must be resistant to various factors, such as lighting conditions, weather changes, and other effects of digital imaging. Inconsistency in predictions under different conditions might result in undesirable outcomes, especially in safety-critical applications such as autonomous driving. To measure the robustness of the CL method against such natural corruptions, we created a dataset by applying fifteen different corruptions, at varying levels of severity (1- least severe to 5- most severe corruption).

The performances on the fifteen corruptions are averaged at each severity level and are shown in Figure 3 (right). DUCA outperforms all other techniques at all severity levels. ER, DER++, and CLS-ER show a fast decline in accuracy as severity increases, while DUCA maintains stable performance throughout. Implicit shape information provides a different perspective of the same data to the model, which helps to generate high-level robust representations. DUCA, along with improved continual learning performance, also exhibits improved robustness to corruption, thus proving to be a better candidate for deployment in real-world applications.

## 6.4 Task-wise Performance

The average accuracy across all tasks does not provide a complete measure of the ability of a network to retain old information while learning new tasks. To better represent the plasticity-stability measure, we report the task-wise performance at the end of each task. After training each task, we measure the accuracy on the test set of each of the previous tasks. Figure 4 reports this for all tasks of *DN4IL*. The last row represents the performance of each task after the training is completed. ER and DER++ show performance degradation on earlier tasks, as the model continues to train on newer tasks. Both perform better than DUCA on the last task, 71.1 and 68.9 respectively, while DUCA has a performance of 61.1.

However, in continual learning settings, the data arrives continuously and the focus is on both retention of old task performance while performing well on the current task. As seen, the accuracy on the first task (real)

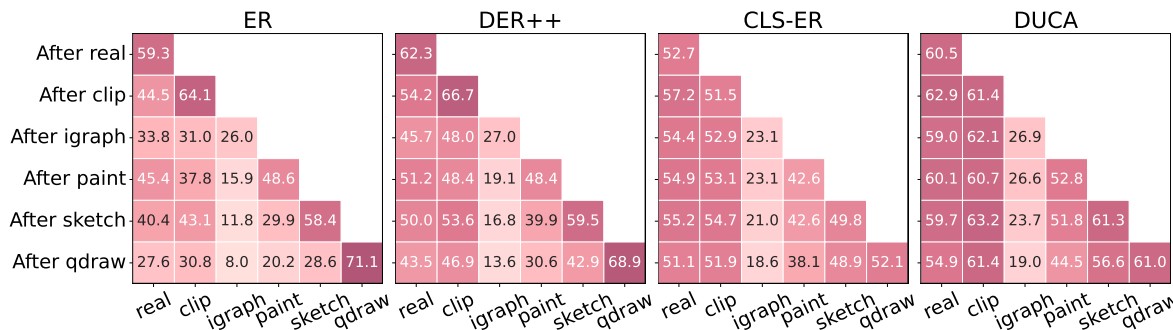

Figure 4: Task-wise performance on *DN4IL* ($|\mathcal{B}|$=500), where each task represents a domain. DUCA shows more retention of old information without compromising much on current accuracy.

Table 2: Analyzing the impact of inductive bias and knowledge sharing on baselines and DUCA. '*' indicates the use of shape as an augmentation. '−X' indicates the removal of component X. '+' refers to concatenation in the channel dimension.

| METHOD | SPECIFICS | SEQ-CIFAR100 | | DN4IL | |
|---|---|---|---|---|---|
| | | 200 | 500 | 200 | 500 |
| ER | ORIGINAL | **21.40** | **28.02** | **26.59** | **31.01** |
| | RGB & SHAPE* | 19.47 | 23.96 | 27.45 | 33.44 |
| DER++ | ORIGINAL | **29.60** | **41.40** | **34.75** | 41.87 |
| | RGB & SHAPE* | 24.40 | 34.30 | 36.55 | 40.99 |
| DUCA | ORIGINAL | **45.38** | **54.27** | **44.23** | **49.32** |
| | −SM (RGB & SHAPE*) | 24.34 | 32.64 | 36.80 | 43.88 |
| | −SM −IBL (SHAPE ONLY) | 18.33 | 21.98 | 27.89 | 31.57 |
| | −SM −IBL (RGB+SHAPE) | 20.57 | 25.20 | 31.52 | 35.68 |
| | −SM −IBL (RGB & SHAPE*) | 19.47 | 23.96 | 27.45 | 33.44 |
| | −IBL (RGB & SHAPE*) | 42.01 | 49.55 | 40.75 | 43.99 |

reduces to 27.6 on ER and 43.5 on *DER++* after training the six tasks (domains), while the DUCA maintains the accuracy of 54.9. Similarly, after the second task, the performance on the first task decreases (44.5 : ER, 54.2 : DER++, 57.2 : CLS-ER and 62.9 : DUCA), but with DUCA the forgetting is lesser. DUCA reports the highest information retention on older tasks, while also maintaining plasticity. For example, CLS-ER shows better retention of old information, but at the cost of plasticity. The last task in CLS-ER shows a lower performance compared to DUCA (52.1 vs. 61.0). The performance of the current task in DUCA is relatively lesser and can be attributed to the stochastic update rate. Therefore, it's essential to recognize that while the shape inductive bias is beneficial to a classification task, the observed high performances are a consequence of how the inductive bias is thoughtfully integrated into the proposed architecture within the framework of continual learning, where both stability and plasticity hold equal importance.

To shed more light on the performance of each of the modules in DUCA, we also provide the performance of the working model and the inductive bias learner, in Appendix Figure 5. The working model shows better plasticity, while DUCA (semantic memory) displays better stability. Overall, all modules in the proposed approach present unique attributes that improve the learning process and improve performance.

## 7 Effect of Inductive Bias and Knowledge Sharing

To assess the impact of the specific inductive bias and various ways of integrating it into the training framework, we conducted supplementary experiments. In these experiments, we introduced two additional baselines, ER and DER++, with the utilization of shape as an augmentation technique. Specifically, we

Table 3: Ablation to analyze the effect of each component of DUCA on Seq-CIFAR10 and *DN4IL*.

| SM | IBL | KS (WM↔IBL) | Seq-CIFAR10 | DN4IL |
|:---:|:---:|:---:|:---:|:---:|
| ✓ | ✓ | ✓ | $\mathbf{70.04}_{\pm 1.07}$ | $\mathbf{44.23}_{\pm 0.05}$ |
| ✓ | ✓ | ✗ | $69.28_{\pm 1.34}$ | $40.35_{\pm 0.34}$ |
| ✓ | ✗ | - | $69.21_{\pm 1.46}$ | $39.76_{\pm 0.56}$ |
| ✗ | ✓ | ✓ | $64.61_{\pm 1.22}$ | $37.33_{\pm 0.01}$ |
| ✗ | ✗ | ✗ | $44.79_{\pm 1.86}$ | $26.59_{\pm 0.31}$ |

included the Sobel filter in the augmentation list, alongside RandomCrop and RandomHorizontalFlip, and proceeded with continual training. The results presented in Table 2 demonstrate that this approach yields inferior performance compared to the baseline models trained solely on RGB images. On DN4IL dataset, the performance is slightly better than baseline, as shape is a more discriminative and important feature in this dataset. Thus the incorporation of shape as an augmentation strategy appears to yield suboptimal feature representations, and is also dependent on the dataset.

We also conduct various ablations on the DUCA framework. Specifically, we perform isolations and exclusions of different elements within DUCA, including IBL and SM. Instead, we subject the base network (DUCA -SM -IBL) to three distinct training conditions: (1) exclusive training on shape images (Shape only), (2) concurrent training on both RGB and shape images (RGB+Shape), and (3) training on RGB images with the incorporation of a shape filter as an augmentation (RGB & Shape*). It is worth noting that shape information contributes valuable global semantic insights that complement the visually rich RGB data, thus emphasizing the necessity of both modalities for achieving enhanced generalization. However, training a single network on both distributions simultaneously may not always yield optimal utilization of this information.

Finally, we train the base model within the framework by excluding one component at a time, namely SM and IBL. We train the working model with but excluding the IBL, while also introducing shape as an augmentation. The results (-IBL) show improvement due to the presence of the semantic memory module. Nevertheless, the best performance is achieved when incorporating shape information through an alternative supervisory network, namely the IBL.

These experiments underscore the critical importance of the specific method used to induce this knowledge, highlighting its pivotal role in enhancing the overall training process. In our pursuit to bridge the distribution gap between RGB and shape images, we have leveraged knowledge distillation as a means of self-teaching, where distinct networks collaboratively engage in mutual learning and knowledge sharing. This approach not only sheds light on the significance of effective knowledge sharing but also offers a promising avenue for improving model performance and generalization in complex (visual) tasks.

## 8 Ablation Study

DUCA architecture comprises multiple components, each contributing to the efficacy of the method. The explicit module has the working model, and the implicit module has semantic memory (SM) and inductive bias learner (IBL). Disentangling different components in the DUCA can provide more insight into the contribution of each of them to the overall performance.

Table 3 reports the ablation study with respect to each of these components on both the Seq-CIFAR10 and *DN4IL* datasets. Considering the more complex *DN4IL* dataset, the ER accuracy without any of our components is 26.59. Adding cognitive bias (IBL) improves performance by 40%. Shape information plays a prominent role, as networks need to learn the global semantics of the objects, rather than background or spurious textural information, to translate performance across domains. Adding the dual-memory component (SM) shows an increase of approximately 49% over the vanilla baseline. Furthermore, the KS between explicit and implicit modules on current experiences also plays a key role in performance gain. Combining both of these cognitive components and, in general, following the multi-module design shows a gain of 66%. A similar trend is seen on Seq-CIFAR10.

## 9    Related Works

Rehearsal-based approaches, which revisit examples from the past to alleviate catastrophic forgetting, have been effective in challenging CL scenarios (Farquhar & Gal, 2018). Experience Replay (ER) (Riemer et al., 2018) methods use episodic memory to retain previously seen samples for replay purposes. DER++ (Buzzega et al., 2020) adds a consistency loss on logits, in addition to the ER strategy. In situations where memory limitations impose constraints on buffer size, such as in edge devices, it has been observed that rehearsal-based methods are susceptible to overfitting on the data stored in the buffer (Bhat et al., 2022). To address this, $CO^2L$ (Cha et al., 2021) uses contrastive learning from the self-supervised learning domain to generate transferable representations. ER-ACE (Caccia et al., 2021) targets the representation drift problem in online CL and develops a technique to use separate losses for current and buffer samples. All of these methods limit the architecture to a single stand-alone network, contrary to the biological workings of the brain.

CLS-ER (Arani et al., 2022) proposed a multi-network approach that emulates fast and slow learning systems by using two semantic memories, each aggregating weights at different times. Though CLS-ER utilizes the multi-memory design, sharing of different kinds of knowledge is not leveraged, and hence presents a method with limited scope. DUCA digresses from the standard architectures and proposes a multi-module design that is inspired by cognitive computational architectures. It incorporates multiple submodules, each sharing different knowledge to develop an effective continual learner that has better generalization and robustness.

## 10    Conclusion

We introduced a novel framework for continual learning that incorporates concepts inspired by cognitive architectures, high-level cognitive biases, and the multi-memory system. *Dual Cognitive Architecture (DUCA)*, includes multiple subsystems with dual knowledge representation. DUCA designed a dichotomy of explicit and implicit modules in which information is selected, maintained, and shared with each other to enable better generalization and robustness. DUCA outperformed on Seq-CIFAR10 and Seq-CIFAR100 on the Class-IL setting. In addition, it also showed a significant gain in the more realistic and challenging GCIL setting. Through different analyses, we showed a better plasticity-stability balance.

Furthermore, shape prior and knowledge consolidation helps to learn more generic solutions, indicated by the reduced problem of task recency bias and greater robustness against natural corruptions. Furthermore, we introduced a challenging domain-IL dataset, *DN4IL*, with six disparate domains. The significant improvement of DUCA on this complex distribution shift demonstrates the benefits of shape context, which helps the network to converge on a generic solution, rather than a simple texture-biased one. The objective of this work was to develop a framework that incorporates elements of cognitive architecture to mitigate the forgetting problem and enhance generalization and robustness.

## 11    Future Work

Here, we delve into the potential for extending our current research, acknowledging its applicability to a diverse range of modalities. The adaptability of DUCA serves as a robust foundation for further exploration. As we broaden the scope of DUCA beyond the image domain, an essential consideration is the identification of pertinent inductive biases tailored to the specific modality in question. For example, when venturing into the audio domain, a promising avenue involves the utilization of spectrogram representations. These representations effectively convert audio waveforms into visual data, encompassing both frequency and time-domain information. The integration of phonemes, the fundamental units of spoken language, holds the potential to enhance DUCA's effectiveness in tasks such as speech understanding, speaker identification, and language processing.

The collaboration between the core DUCA architecture and modality-specific inductive biases creates a synergy that drives knowledge sharing and learning capabilities. This collaborative architecture yields more generic and robust representations, substantially enhancing overall performance. Furthermore, the gradual accumulation of semantic memory emerges as a valuable asset, particularly in scenarios involving the contin-

uous influx of data from various modalities. It mitigates the risk of forgetting and empowers the framework to maintain its adaptability over time.

These modality-specific adaptations, guided by the intrinsic principles and mechanisms of DUCA, open the door to exciting future directions. They offer the potential to advance lifelong learning and adaptability in a multitude of domains. We anticipate that our preliminary work will serve as a cornerstone for future research endeavors, including investigations into various cognitive biases and more efficient design methodologies. Ultimately, we hope to pave the way for the advancement of lifelong learning methods for ANNs.

### Acknowledgments

The research was conducted when all the authors were affiliated with Advanced Research Lab, NavInfo Europe, The Netherlands.

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

# A DUCA Algorithm

---

**Algorithm 1** Dual Cognitive Architecture (DUCA)

---
1: **Input:** Dataset $\mathcal{D}_t$, Buffer $\mathcal{B}$
2: **Initialize:** Three networks: Encoder and classifier $f$ parameterized by $\theta_{WM}$, $\theta_{SM}$, and $\theta_{IBL}$
3: **for** all tasks $t \in 1, 2...T$ **do**
4:     Sample mini-batch: $(x_c, y_c) \sim \mathcal{D}_t$
5:     Extract shape images: $x_{c_s} = \mathbb{B}(x_c)$ where $\mathbb{B}$ is a Sobel filter
6:     $\mathcal{L}_{Sup_{WM}} = \mathcal{L}_{CE}(f(x_c; \theta_{WM}), y_c)$
7:     $\mathcal{L}_{Sup_{IBL}} = \mathcal{L}_{CE}(f(x_{c_s}; \theta_{IBL}), y_c)$
8:     **if** $\mathcal{B} \neq \emptyset$ **then**
9:         Sample mini-batch: $(x_b, y_b) \sim \mathcal{B}$
10:         Extract shape images: $x_{b_s} = \mathbb{B}(x_b)$
11:         Calculate the supervised loss:
12:         $\mathcal{L}_{Sup_{WM}} \mathrel{+}= \mathcal{L}_{CE}(f(x_b; \theta_{WM}), y_b)$
13:         $\mathcal{L}_{Sup_{IBL}} \mathrel{+}= \mathcal{L}_{CE}(f(x_{b_s}; \theta_{IBL}), y_b)$
14:         Knowledge sharing from semantic memory to working model and inductive bias learner:
15:         $\mathcal{L}_{KS_{WM}} = \mathbb{E}\|f(x_b; \theta_{SM}) - f(x_b; \theta_{WM})\|_2^2$
16:         $\mathcal{L}_{KS_{IBL}} = \mathbb{E}\|f(x_b; \theta_{SM}) - f(x_{b_s}; \theta_{IBL})\|_2^2$
17:     Bidirectional knowledge sharing between working model and inductive bias learner:
18:     $\mathcal{L}_{biKS} = \mathop{\mathbb{E}}\limits_{x \sim D_t \cup \mathcal{B}} \|f(x; \theta_{WM}) - f(x_s; \theta_{IBL})\|_2^2$
19:     Calculate total loss:
20:     $\mathcal{L}_{WM} = \mathcal{L}_{Sup_{WM}} + \lambda(\mathcal{L}_{biKS} + \mathcal{L}_{KS_{WM}})$
21:     $\mathcal{L}_{IBL} = \mathcal{L}_{Sup_{IBL}} + \gamma(\mathcal{L}_{biKS} + \mathcal{L}_{KS_{IBL}})$
22:     Update both working model and inductive bias learner: $\theta_{WM}, \theta_{IBL}$
23:     Stochastically update semantic memory:
24:     Sample $s \sim U(0, 1)$;
25:     **if** $s < r$ **then**
26:         $\theta_{SM} = d \cdot \theta_{SM} + (1 - d) \cdot \theta_{WM}$
27:     Update memory buffer $\mathcal{B}$
28: **Return:** model $\theta_{SM}$

---

# B Effect of Task Sequence

Figure 5 presents the task-wise performance of all the three networks in the DUCA architecture, on *DN4IL* dataset. Semantic memory helps to retain information by maintaining high accuracy on older tasks and is more stable. The performance of the current task is relatively lower than that of the working model and could be due to the stochastic update rate of this model. The working model has better performance on new tasks and is more plastic. Inductive bias leaner is evaluated on the transformed data (shape) and also achieves a balance between plasticity and stability. In general, all modules in our proposed method present unique attributes that improve the learning process by improving performance and reducing catastrophic forgetting. Table 4 reports the results of CIFAR100 on different numbers of tasks. Even when the number of tasks increases, our method consistently improves over the baselines.

Table 4: Comparison on Seq-CIFAR100 dataset for different number of tasks on 500 buffer size.

| METHOD | 5-TASKS | 10-TASKS | 20-TASKS |
|---|---|---|---|
| ER | $28.02_{\pm 0.31}$ | $21.49_{\pm 0.47}$ | $16.52_{\pm 0.86}$ |
| DER++ | $41.40_{\pm 0.96}$ | $36.20_{\pm 0.52}$ | $22.25_{\pm 5.87}$ |
| DUCA | $\mathbf{53.23}_{\pm 1.62}$ | $\mathbf{41.09}_{\pm 0.72}$ | $\mathbf{33.60}_{\pm 0.25}$ |

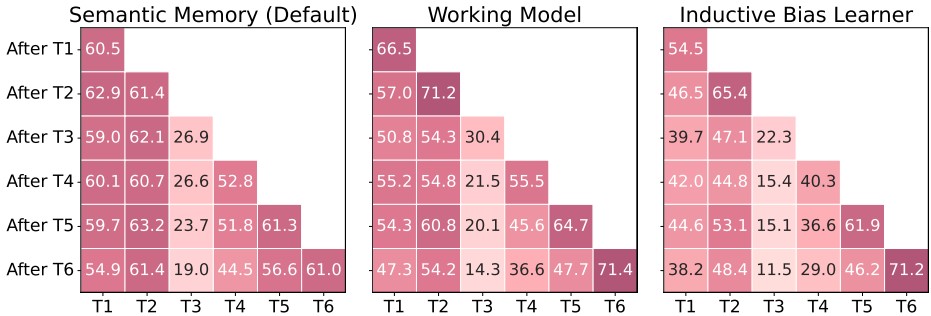

Figure 5: Task probability analysis of all DUCA components on *DN4IL* dataset with 500 buffer size. Semantic memory displays better stability while the working model displays better plasticity.

Table 5: Comparison on R-MNIST dataset with different buffer sizes.

| METHOD | 200 | 500 |
|--------|-----|-----|
| ER | $85.01_{\pm1.90}$ | $88.91_{\pm1.44}$ |
| DER++ | $90.43_{\pm1.87}$ | $92.77_{\pm1.05}$ |
| CLS-ER | $92.26_{\pm0.18}$ | $94.06_{\pm0.07}$ |
| DUCA | $\mathbf{93.53}_{\pm0.21}$ | $\mathbf{94.98}_{\pm0.04}$ |

# C  *Evaluation on R-MNIST*

The Rotated MNIST (R-MNIST) dataset is a variant of the MNIST dataset, which is widely used for domain-incremental learning in continual learning. In the R-MNIST dataset, the digits are rotated by fixed degrees. This rotation introduces variations in the orientation of the digits, making it a more challenging task for machine learning algorithms to accurately classify them. It tests the robustness and adaptability of models, as they need to recognize and classify the digits regardless of their orientation. The results, as presented in Table 5, indicate that DUCA performs better than other methods in this domain-incremental setting.

# D  Setting and Datasets

We evaluate all methods in different CL settings. Van de Ven & Tolias (2019) describes three different settings based on increasing difficulty: task-incremental learning (Task-IL), domain-incremental learning (Domain-IL), and class-incremental learning (Class-IL). In Class-IL, each new task consists of novel classes, and the network must learn both new classes while retaining information about the old ones. Task-IL is similar to Class-IL but assumes that task labels are accessible in both training and inference. In Domain-IL, the classes remain the same for each task, but the distribution varies for each task. We report the results for all three settings on the relevant datasets. CLass-IL is the most complex setting of the three and is widely studied. However, contemporary research tends to adopt simplifications when exploring CLass-IL setting, such as the assumption of the same number of classes across different tasks, the absence of reappearance of classes, and the sample distribution per class being well balanced. In Generalized Class-IL (GCIL) Mi et al. (2020), the number of classes in each task is not fixed, and the classes can reappear with varying sample sizes. GCIL samples the number of classes and samples from a probabilistic distribution. The two variations are Uniform (fixed uniform sample distribution over all classes) and Longtail (with class imbalance). We report results on all three settings: Task-IL, Domain-IL, and CLass-IL. Furthermore, we also consider the GCIL setting for one of the datasets as an additional evaluation setting. All reported results are averaged over three random seeds.

Table 6: Search ranges for tuning hyperparameters.

| METHOD | HYPERPARAM | SEARCH RANGE | METHOD | HYPERPARAM | SEARCH RANGE |
|---|---|---|---|---|---|
| ER | $lr$ | [0.01, 0.03, 0.1, 0.5] | | $lr$ | [0.01, 0.03, 0.1] |
| DER++ | $lr$ | [0.01, 0.03, 0.1] | | $\lambda$ | [0.1, 0.2, 0.3] |
| | $\alpha$ | [0.1, 0.2, 0.5] | CLS-ER | $r_p$ | [0:1:0.1] |
| | $\beta$ | [0.5, 1.0] | | $r_s$ | [0:1:0.1] |
| CO$^2$L | $lr$ | [0.01, 0.03, 0.1] | | $\alpha_p$ | [0.99,0.999] |
| | $\tau$ | [0.01, 0.1, 0.5] | | $\alpha_s$ | [0.99, 0.999] |
| | $k$ | [0.2, 0.5] | | $lr$ | [0.01, 0.03, 0.1] |
| | $k^*$ | [0.01, 0.05] | | $r$ | [0:1:0.1] |
| | $e$ | [100, 150] | DUCA | $\lambda$ | [0.01, 0.1] |
| ER-ACE | $lr$ | [0.01, 0.03, 0.1, 0.5] | | $\gamma$ | [0.01, 0.1] |

## E    Hyperparameters

For the settings and datasets for which the results are not available in the original papers, using their original codes, we conducted a hyperparameter search for each of the new settings. To this end, we utilize a small validation set from the training set to tune the hyperparameters. We adopted this tuning scheme because it aligns with the approach employed in the baseline methods with which we conducted comparisons (Buzzega et al., 2020; Cha et al., 2021; Caccia et al., 2021; Arani et al., 2022).

For Seq-CIFAR10, the results are taken from the original articles (Buzzega et al., 2020; Cha et al., 2021; Caccia et al., 2021; Arani et al., 2022). For the other datasets (and for each buffer size), we ran a grid search over the hyperparameters reported in the paper. For Seq-CIFAR100 and GCIL-CIFAR100, we based the search range using Seq-CIFAR10 hyperparameters as a reference point. The search ranges are reported in Table 6. Once we find the right hyperparameter, we train the whole training set and report the test set accuracy.

DN4IL dataset is more complex compared to the CIFAR versions and includes images of larger sizes. Hence, we consider the Seq-TinyImagenet hyperparameters in the respective paper as a reference point for further tuning. The learning rate $lr$, the number of epochs, and the batch size are similar across the datasets. The ema update rate $r$ is lower for more complex datasets, as shown in CLS-ER. $r$ is chosen in the range of $[0.01, 0.1]$ with a step size of 0.02 for CLS-ER and DUCA. The different hyperparameters chosen for the baselines, after tuning, are reported in Table 7.

The different hyperparameters chosen for DUCA are shown in Table 8. Hyperparameters: $lr$, batch size, number of epochs and decay factor are uniform across all datasets. The stochastic update rate is similar to those of CLS-ER. The hyperparameters are stable across settings and datasets and also complement each other. The loss balancing weights ($\lambda$ and $\gamma$ respectively) are also similar across all datasets. Therefore, DUCA does not require extensive fine-tuning across different datasets and settings.

## F    Model Complexity

We discuss the computational complexity aspect of our proposed method. DUCA involves three networks during training; however, in inference, only one network is used (SM module). Therefore, for inference purposes, the MAC count, the number of parameters, and the computational capacity remain the same as in the other single-network methods.

The training cost requires three forward passes, as it consists of three different modules. ER, DER++, CO$^2$L and ER-ACE have a single network. CLS-ER also has three networks and therefore requires 3 forward passes. DUCA has training complexity similar to that of CLS-ER; however, it outperforms CLS-ER in all provided metrics. On the memory front, similar to all methods, we save memory samples based on the memory budget allotted (200 and 500 in the experiments). There are no additional memory requirements, as we do not save

Table 7: Selected hyperparameters for all baselines.

| DATASET | $|\mathcal{B}|$ | METHOD | HYPERPARAMETERS |
|---|---|---|---|
| SEQ-CIFAR100 | 200 | ER | $lr$=0.1 |
| | | DER++ | $lr$=0.03, $\alpha$=0.1, $\beta$=0.5 |
| | | CO$^2$L | $lr$:0.5, $\tau$:0.5, $\kappa$:0.2, $\kappa^*$:0.01, $e$:100 |
| | | ER-ACE | $lr$=0.01 |
| | | CLS-ER | $lr$=0.1 $\lambda$=0.15, $r_p$=0.1, $r_s$=0.05, $\alpha_p$=0.999, $\alpha_s$=0.999 |
| | 500 | ER | $lr$=0.1 |
| | | DER++ | $lr$=0.03, $\alpha$=0.1, $\beta$=0.5 |
| | | CO$^2$L | $lr$:0.5, $\tau$:0.5, $\kappa$:0.2, $\kappa^*$:0.01, $e$:100 |
| | | ER-ACE | $lr$=0.01 |
| | | CLS-ER | $lr$=0.1 $\lambda$=0.15, $r_p$=0.1, $r_s$=0.05, $\alpha_p$=0.999, $\alpha_s$=0.999 |
| GCIL-CIFAR100 | 200 | ER | $lr$=0.1 |
| | | DER++ | $lr$=0.03, $\alpha$=0.5, $\beta$=0.1 |
| | | ER-ACE | $lr$=0.1 |
| | | CLS-ER | $lr$=0.1 $\lambda$=0.1, $r_p$=0.7, $r_s$=0.6, $\alpha_p$=0.999, $\alpha_s$=0.999 |
| | 500 | ER | $lr$=0.1 |
| | | DER++ | $lr$=0.03, $\alpha$=0.2, $\beta$=0.1 |
| | | ER-ACE | $lr$=0.1 |
| | | CLS-ER | $lr$=0.1 $\lambda$=0.1, $r_p$=0.7, $r_s$=0.6, $\alpha_p$=0.999, $\alpha_s$=0.999 |
| DN4IL | 200 | ER | $lr$=0.1 |
| | | DER++ | $lr$=0.03, $\alpha$=0.1, $\beta$=1.0 |
| | | CLS-ER | $lr$=0.05 $\lambda$=0.1, $r_p$=0.08, $r_s$=0.04, $\alpha_p$=0.999, $\alpha_s$=0.999 |
| | 500 | ER | $lr$=0.1 |
| | | DER++ | $lr$=0.03, $\alpha$=0.5, $\beta$=0.1 |
| | | CLS-ER | $lr$=0.05 $\lambda$=0.1, $r_p$=0.08, $r_s$=0.05, $\alpha_p$=0.999, $\alpha_s$=0.999 |

Table 8: Selected hyperparameters for DUCA across different settings. The learning rate is set to 0.03, batch size to 32, and epochs to 50 respectively for all the datasets. The decay factor $d$ is always set to 0.999.

| DATASET | $|\mathcal{B}|$ | $r$ | $\lambda$ | $\gamma$ | DATASET | $|\mathcal{B}|$ | $r$ | $\lambda$ | $\gamma$ |
|---|---|---|---|---|---|---|---|---|---|
| SEQ-CIFAR10 | 200 | 0.2 | 0.1 | 0.1 | GCIL-CIFAR100 | 200 | 0.09 | 0.1 | 0.01 |
| | 500 | 0.2 | 0.1 | 0.1 | | 500 | 0.09 | 0.1 | 0.01 |
| SEQ-CIFAR100 | 200 | 0.1 | 0.1 | 0.01 | DN4IL | 200 | 0.06 | 0.1 | 0.01 |
| | 500 | 0.06 | 0.1 | 0.01 | | 500 | 0.08 | 0.1 | 0.01 |

any extra information (such as logits in DER++) to be used later in our objectives. Additionally, there are no additional data requirements. The number of parameters is reported in Table 9.

We considered the human brain, or cognition, as the most intelligent agent and wanted to incorporate some of the underlying workings into the neural network architecture. Therefore, our goal was to design a framework inspired by elements of cognitive architecture that reduce forgetting while exhibiting better generalization and robustness. Our work is a preliminary attempt to incorporate elements based on cognitive architecture into the CL algorithm to gauge the gain in reducing forgetting. We hope the promising results result in future work that explores different kinds of cognitive biases and interactions and moves towards more efficient designs.

Table 9: Comparison of the number of parameters for different methods using ResNet18 architecture and trained on SeqCIFAR100 dataset.

| METHOD | TRAINING | INFERENCE |
|--------|----------|-----------|
| ER | 11 | 11 |
| DER++ | 11 | 11 |
| CO$^2$L | 23 | 11 |
| ER-ACE | 11 | 11 |
| CLS-ER | 33 | 11 |
| DUCA | 33 | 11 |

---

**Algorithm 2** Sobel Algorithm - Shape Extraction

---

1: Up-sample the images to twice the original size: $x_{rgb} = \text{us}(x_{rgb})$
2: Reduce noisy edges: $x_g = \text{Gaussian\_Blur}(x_{rgb}, kernel\_size = 3)$
3: Get Sobel kernels: $S_x = \begin{bmatrix} -1 & 0 & +1 \\ -2 & 0 & +2 \\ -1 & 0 & +1 \end{bmatrix}$ and $S_y = \begin{bmatrix} -1 & -2 & -1 \\ 0 & 0 & 0 \\ +1 & +2 & +1 \end{bmatrix}$
4: Apply Sobel kernels: $x_{dx} = x_g * S_x$ and $x_{dy} = x_g * S_y$ ($*$ is 2-D convolution operation)
5: The edge magnitude: $x_{shape} = \sqrt{x_{dx}^2 + x_{dy}^2}$
6: Down-sample to original image size: $x_{shape} = \text{ds}(x_{shape})$

---

## G   Inductive Bias

The shape extraction is performed by applying a filter on the input image. Multiple filters (such as Canny (Ding & Goshtasby, 2001), Prewitt) were considered, but the Sobel filter (Sobel & Feldman, 1968) was chosen because it produces a more realistic output by being precise and also smoothing the edges (Gowda et al., 2022); see Algorithm 2. Figure 6 shows a few examples of applying the Sobel operator on the original RGB images. The Sobel output is fed to the IBL model.

The DUCA framework can be extended to other domains by selecting appropriate inductive biases relevant to those domains. For example, in the audio domain, one viable option involves leveraging spectrogram representations, which transform audio waveforms into visual data capturing both frequency and time-domain information. Alternatively, the incorporation of phonemes, representing the fundamental sound units in spoken language and carrying essential linguistic information, can enhance the framework's capabilities for speech understanding, speaker identification, and language processing. Additionally, the extraction of pitch and timbre features proves valuable for tasks like melody extraction, intonation analysis, and various music-related applications, providing insights into the acoustic characteristics of audio signals.

## H   *DN4IL Dataset*

We introduce a new dataset for the Domain-IL setting. It is a subset of the standard DomainNet dataset (Peng et al., 2019) used in domain adaptation. It consists of six different domains: real, clipart, infograph, painting, quickdraw, and sketch. The shift in distribution between domains is challenging. A few examples can be seen in Figure 7.

Each domain includes 345 classes, and the overall dataset consists of ∼59000 samples. The classes have redundancy, and also evaluating on the whole dataset can be computationally expensive for CL settings. Therefore, we create a subset by grouping semantically similar classes into 20 supercategories (considering the class overlap between other standard datasets can also facilitate OOD analysis). Each super category has five classes each, which results in a total of 100 classes. The specifications of the classes are given in Table 11. The dataset consists of 67080 training images and 19464 test images. The image size for all experiments is chosen as 64×64 (the normalize transform is not applied in augmentations).

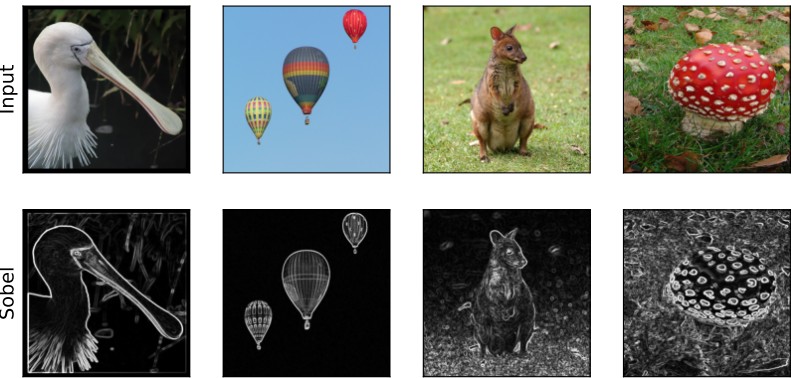

Figure 6: Visual examples of the shape images using Sobel operator

Table 10: Accuracy on the proposed *DN4IL* dataset for the Domain-IL setting. DUCA shows a significant improvement in all disparate and challenging domains.

| $|\mathcal{B}|$ | METHOD | REAL | CLIPART | INFOGRAPH | PAINTING | SKETCH | QUICKDRAW | ACC |
|---|---|---|---|---|---|---|---|---|
| - | JOINT | | | | | | | 59.93±1.07 |
| | SGD | 9.98±0.54 | 19.97±0.31 | 2.32±0.20 | 6.58±0.34 | 14.91±0.04 | 71.23±0.17 | 20.83±0.24 |
| 200 | ER | 20.08±0.45 | 26.37±0.35 | 5.56±0.39 | 13.92±0.91 | 23.69±1.54 | 69.95±0.56 | 26.59±0.31 |
| | DER++ | 33.66±1.65 | 37.24±0.64 | 9.80±0.63 | 24.16±1.17 | 34.37±2.00 | 69.26±0.79 | 34.75±0.87 |
| | CLS-ER | 45.53±0.88 | 49.17±1.12 | 15.79±0.48 | 35.80±0.64 | 48.03±0.85 | 54.40±1.25 | 40.83±1.07 |
| | DUCA | 47.52±0.25 | 54.69±0.10 | 15.70±0.33 | 37.54±0.30 | 51.98±0.96 | 58.80±0.18 | **44.23**±0.05 |
| 500 | ER | 27.54±0.05 | 31.89±0.93 | 7.89±0.45 | 19.39±1.02 | 28.36±1.35 | 70.96±0.10 | 31.01±0.62 |
| | DER++ | 44.49±1.39 | 46.17±0.35 | 14.01±0.23 | 33.44±0.90 | 43.59±1.11 | 69.53±0.29 | 41.87±0.63 |
| | CLS-ER | 49.85±0.88 | 51.41±0.34 | 18.17±0.08 | 37.94±0.94 | 49.02±1.57 | 55.63±0.71 | 43.41±0.80 |
| | DUCA | 54.77±0.15 | 60.37±0.75 | 19.35±0.39 | 44.50±0.43 | 56.34±0.53 | 60.61±1.73 | **49.32**±0.23 |

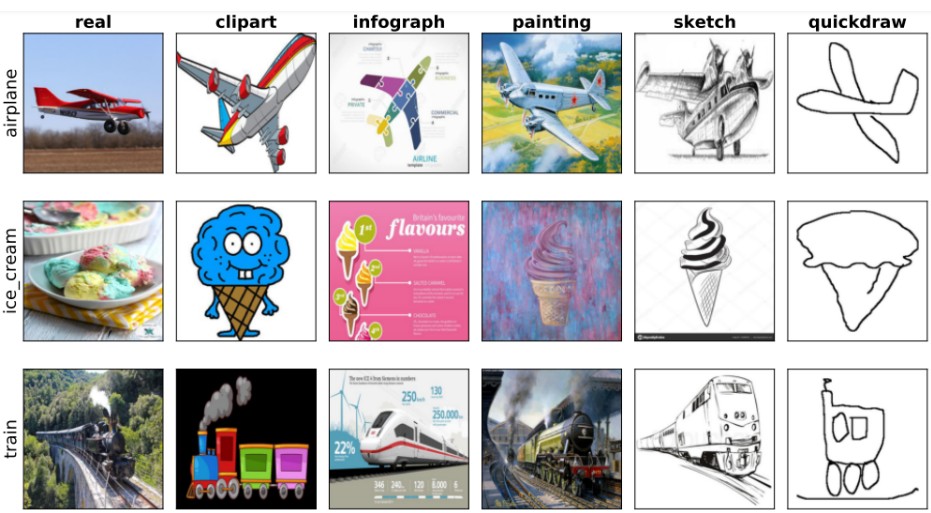

Figure 7: Visual examples of *DN4IL* dataset

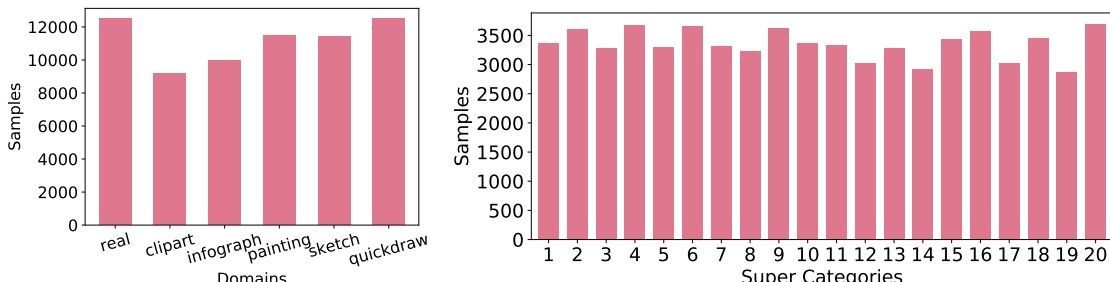

Figure 8: Number of samples per domain and per supercategory in *DN4IL* dataset.

Table 11: Details on supercategory and classes in *DN4IL* dataset.

| | SUPERCATEGORY | CLASS | | | | |
|---|---|---|---|---|---|---|
| 1 | SMALL ANIMALS | MOUSE | SQUIRREL | RABBIT | DOG | RACCOON |
| 2 | MEDIUM ANIMALS | TIGER | BEAR | LION | PANDA | ZEBRA |
| 3 | LARGE ANIMALS | CAMEL | HORSE | KANGAROO | ELEPHANT | COW |
| 4 | AQUATIC MAMMALS | WHALE | SHARK | FISH | DOLPHIN | OCTOPUS |
| 5 | NON-INSECT INVERTEBRATES | SNAIL | SCORPION | SPIDER | LOBSTER | CRAB |
| 6 | INSECTS | BEE | BUTTERFLY | MOSQUITO | BIRD | BAT |
| 7 | VEHICLE | BUS | BICYCLE | MOTORBIKE | TRAIN | PICKUP__TRUCK |
| 8 | SKY-VEHICLE | AIRPLANE | FLYING__SAUCER | AIRCRAFT__CARRIER | HELICOPTER | HOT__AIR__BALLOON |
| 9 | FRUITS | STRAWBERRY | BANANA | PEAR | APPLE | WATERMELON |
| 10 | VEGETABLES | CARROT | ASPARAGUS | MUSHROOM | ONION | BROCCOLI |
| 11 | MUSIC | TROMBONE | VIOLIN | CELLO | GUITAR | CLARINET |
| 12 | FURNITURE | CHAIR | DRESSER | TABLE | COUCH | BED |
| 13 | HOUSEHOLD ELECTRICAL DEVICES | CLOCK | FLOOR__LAMP | TELEPHONE | TELEVISION | KEYBOARD |
| 14 | TOOLS | SAW | AXE | HAMMER | SCREWDRIVER | SCISSORS |
| 15 | CLOTHES & ACCESSORIES | BOWTIE | PANTS | JACKET | SOCK | SHORTS |
| 16 | MAN-MADE OUTDOOR | SKYSCRAPER | WINDMILL | HOUSE | CASTLE | BRIDGE |
| 17 | NATURE | CLOUD | BUSH | OCEAN | RIVER | MOUNTAIN |
| 18 | FOOD | BIRTHDAY__CAKE | HAMBURGER | ICE__CREAM | SANDWICH | PIZZA |
| 19 | STATIONARY | CALENDAR | MARKER | MAP | ERASER | PENCIL |
| 20 | HOUSEHOLD ITEMS | WINE__BOTTLE | CUP | TEAPOT | FRYING__PAN | WINE__GLASS |

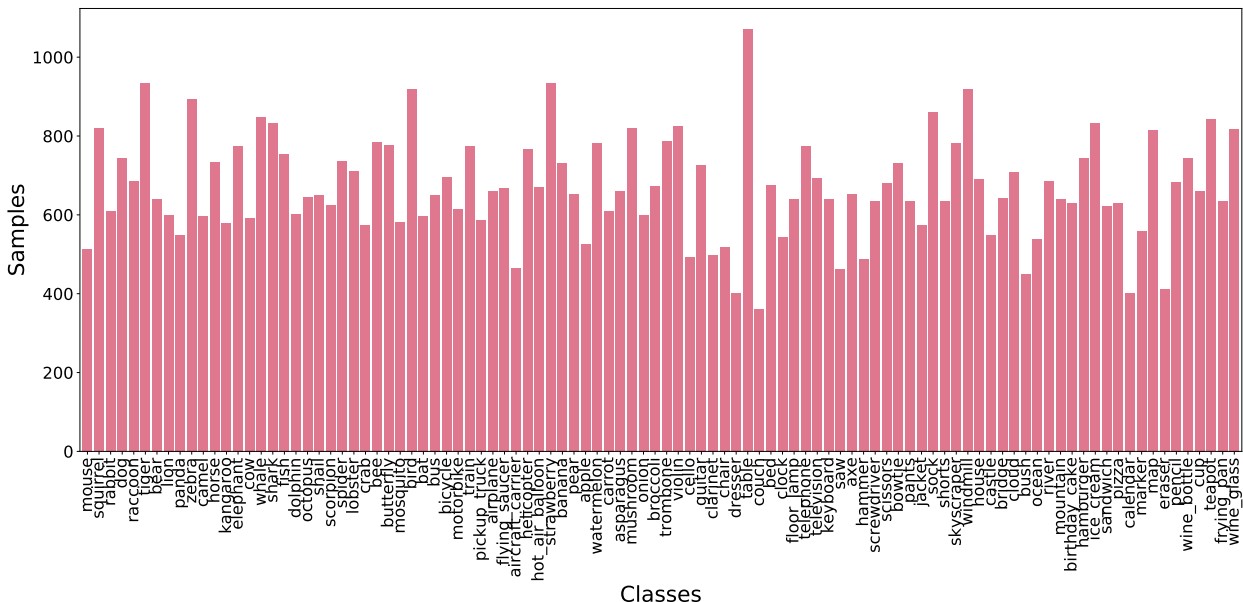

Figure 9: Number of overall samples per class in *DN4IL* dataset.

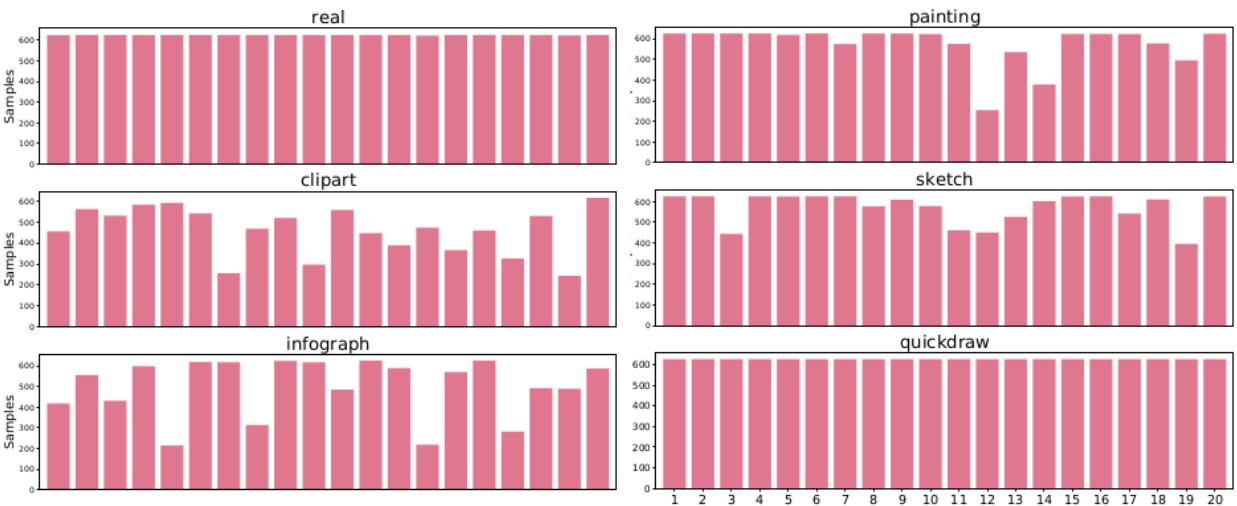

Figure 10: Number of samples per supercategory for each domain in *DN4IL* dataset.

