# OpenReview forum: "Dual Cognitive Architecture: Incorporating Biases and Multi-Memory Systems for Lifelong Learning"
_TMLR — Accepted by TMLR_

### Review · Reviewer_3ogS · 2023-06-03

**Summary Of Contributions:**

The author propose a novel method for continual learning  (CL), which is both rehearsal- and  architecture-based.

The method consists in augmenting the main network with a"semantic memory" (a running average of the main network updated at random times) and an "inductive bias" system (a network  trained on edge-detected versions of the input images, presumably emphasizing shape information). The subsystems are trained on both current examples and previous examples, stored in a replay buffer. IIIUC the outputs of the various networks are constrained to be similar by MSE losses (Eqs. 3,4). The method is tested on various continual learning  problems, representing Task-, Domain- and Class-incremental  learning, and shows significant benefits over existing approaches.

In addition, a new dataset for domain-incremental learning (essentially curated form a larger existing dataset) is introduced.

**Audience:**

Yes

**Broader Impact Concerns:**

I do not see any broader impact concerns.

**Claims And Evidence:**

Yes

**Requested Changes:**

- Please add number of trainable parameters and if possible wallclock training time for the various approaches.

- At the very least, explain the benefit of adding  a dedicated shape-based  network to  the system, rather than merely augmenting the main dataset with edge-detected images. Actual experiments would be best, if  at all possible.

- Section 2.4, second paragraph: what  are the "encoder" and the "classifier"? This does not seem to have been discussed beforehand.

- Last paragraph of p. 1: fix the second sentence.

- Last paragraph of p.3: "deferentially" -> "differentially".

**Strengths And Weaknesses:**

Strengths:

- The method seems novel.

- It seems to work well.

- The new dataset is potentially interesting.

Weaknesses:

There doesn't seem to be any deadly flaw in the proposed method, but some question marks.

- From the results,  it seems that the addition of the shape-based model (trained on edge-detection images) was the main factor in the approach's superior performance (see especially  section 6.5 on ablations). Thus, the main  result of the paper may simply be that image-based tasks benefit from a bias towards shapes. This may be interesting, but it may also seem somewhat orthogonal to  the actual problem of lifelong learning / CL.

- While the authors address the complexity of the various methods, they do not address the number of trainable parameters (unless I missed it). Are the comparison between models with equal numbers of trainable parameters and wallclock time?

- The system includes a dedicated  network trained on edge-detected images. This plays a large part in the model's performance. But is this improvement caused by the architectural innovation, or simply the inclusion of edge-detected /shape-emphasized images  in the data? An alternative would be to augment main training set with edge-detected images, using only the main network and the "semantic" (slow-update)  network in the  model, without including a  dedicated "inductive bias" network. How would this simpler method compare with the proposed approach, all  other things (training time, number of parameters, etc.) being equal?

---

> ### Author Response · Authors · 2023-09-06
> **Response to Reviewer 3ogS**
>
> We express our appreciation to the reviewer for offering us insightful comments and constructive feedback. We have furnished our responses below, and should any further inquiries arise, please feel free to reach out to us.
>
> >  the main result of the paper may simply be that image-based tasks benefit from a bias towards shapes. This may be interesting, but it may also seem somewhat orthogonal to the actual problem of lifelong learning / CL
>
> The aim is to provide an inductive bias towards shape, which helps in capturing global semantics and improving generalization and also help tackle the challenges of plasticity, stability, and knowledge retention in continual learning scenarios.
>
> Further, we also provide more baselines by using shape as an augmentation in the table below (and also in Table 4 in Appendix). We add shape as an augmentation instead of using an inductive bias learner (IBL) network.. The results show that while shape is beneficial, the way it is incorporated to the training scheme is of the utmost importance.
>
> We also evaluate task-wise performance, for instance in Figure3, which shows the task-wise performance on DN4IL dataset, we see that after the first task, DER++ (62.3)  has the highest performance and not DUCA (60.5). But after all 6 tasks, the performance of task1 reduces to 43.5 in DER++, while DUCA has an accuracy of 54.9, thus showing more retention ability. Thus, the incorporation of shape knowledge proves instrumental in directing the model's attention towards important features within each task, ultimately leading to an overall enhancement in performance. In continual learning the focus is on overall retention of old task performance while also performing well on the current task and DUCA achieves this while maintaining a better balance between plasticity and stability.
>
>
> > While the authors address the complexity of the various methods, they do not address the number of trainable parameters (unless I missed it). Are the comparison between models with equal numbers of trainable parameters and wallclock time?
>
> We tabulate the number of parameters in the table below (and also Table9 in Appendix).
>
> | Method | Training | Inference |
> |:-|---|---:|
> | ER | 11 | 11 |
> | DER++ | 11 | 11 |
> | CO2L | 23 | 11 |
> | ER-ACE | 11 | 11 |
> | CLS-ER | 33 | 11 |
> | DUCA | 33 | 11 |
>
> Further, in terms of other additional requirements needed by each method, DER utilizes saved logits, DER++ saves logits and labels to perform an additional operation, and CO2L stores previous models and also utilizes more augmentations than others. And, it is prominent to note that even with a smaller buffer size of 200, DUCA outperforms other methods that utilize a larger memory of 500, as shown in Table 1 for Seq-CIFAR100 and GCIL-CIFAR100. This highlights DUCA's superior performance even in low-memory settings.
>
> > But is this improvement caused by the architectural innovation, or simply the inclusion of edge-detected /shape-emphasized images in the data? An alternative would be to augment main training set with edge-detected images, using only the main network and the "semantic" (slow-update) network in the model, without including a dedicated "inductive bias" network. How would this simpler method compare with the proposed approach, all other things (training time, number of parameters, etc.) being equal?
>
> The novelty of the proposed approach lies not only in the inclusion of shape information but also in how this knowledge is integrated into the architecture. Simply augmenting the main training set with edge-detected or shape-emphasized images may not lead to optimal results. The key aspect of the proposed method is the incorporation of the shape-based model as a subtle supervisory feature in the implicit module. This allows for the learning of better representations by leveraging the self teaching concept, which enables knowledge sharing between the working model and the shape model.
>
> We provide more results to substantiate this. We train DUCA by using RGB images and shape as an augmentation, in two settings (1) without both SM and IBL (-SM -IBL)  (2) without IBL (-IBL)
> - Decreases the results as training with shape as an augmentation may not be the best way to learn the shape information by the working model.
> - Adding SM, improves the results but is still lower than DUCA which has a dedicated shape network to learn better generalization. \
> More ablations are added in Table 4 in the Appendix.
>
>
> | Method | Specifics | Seq- CIFAR100 |  | DN4IL |  |
> |---|---|:---:|:---:|:---:|:---:|
> |  |  | 200 | 500 | 200 | 500 |
> | DUCA | Original | 45.38 | 54.27 | 44.23 | 49.32 |
> |  | -SM -IBL (RGB & Shape*) | 19.47 | 23.96 | 27.45 | 33.44 |
> |  | -IBL (RGB & Shape*) | 42.01 | 49.55 | 40.75 | 43.99 |
>
> > Minor comments about the text
>
> We have incorporated all the modifications into the revised version of the paper, clearly marked in blue. If, by any chance, we have overlooked any changes, please bring them to our attention.

---

### Review · Reviewer_jNRv · 2023-06-08

**Summary Of Contributions:**

Taking inspiration from cognitive science, this paper proposes a new multi-module architecture for continual learning. The proposed “Dual Cognitive Architecture” (or DULA) is made up of a working model (which is similar to a “standard” deep learning architecture), an inductive bias learner (which is trained on pre-processed shape information) and a semantic memory (which essentially is a delayed version of the working model). Several loss terms are introduced to encourage the outputs of these different modules to be comparable to each other. Using variants of the CIFAR10 and CIFAR100 datasets, the authors test the performance of the new architecture on task- and class-incremental learning settings; and for the domain-incremental learning setting a new dataset DN4IL is proposed, which is a curated subset of DomainNet. In all reported experiments, DUCA performs better than several established replay-based continual learning methods (such as DER++ and Co2L).

**Audience:**

Yes

**Broader Impact Concerns:**

No concerns in this regard.

**Claims And Evidence:**

No

**Requested Changes:**

I believe the main change that should be made is providing evidence that, and ideally also insights why, the various proposed aspects of DUCA have benefits that are specific for continual learning.

Otherwise, please refer to the various raised issues and suggestions under “Strengths and Weaknesses”, also for more details on the requested change above.

**Strengths And Weaknesses:**

I think the main strength (or perhaps “selling point” is a better word) of this paper is the strong performance of the proposed method DUCA relative to established replay-based CL methods such as DER++ and Co2L. The suggestion that using shape information can have a specific benefit for continual learning is intriguing as well. Finally, I think that the proposed DN4IL dataset can be a useful contribution to the continual learning literature, albeit somewhat incremental.

However, that the strong performance of DUCA is the main selling point, is in my opinion also an important weakness of this paper. The goal of continual learning research is not to achieve “state-of-the-art” performance on variants of Split CIFAR; rather the goal is to gain insights into how to build successful CL algorithms (and to eventually apply those to practical problems, although that goal is still distant). Controlled benchmarking experiments on variants of Split CIFAR could help towards this goal, but by themselves such experiments are of limited value, especially if there are concerns regarding how controlled they are.

Perhaps the authors would argue that an insight from their paper is that inductive bias (or more specifically, a bias towards using shape information of images) can be beneficial for continual learning, but I’m afraid I do not think the current version of the paper satisfactorily demonstrates this. The authors show that using shape information can boost CL performance. However, as it is already known that using shape information can boost image classification (e.g., Gheiros et al, 2019 ICLR), it is not clear whether the use of shape information has benefits for CL on top of the general benefits it can have for image classification. I think the usage of shape information, and in particular the way this paper proposes to use shape information, *could* carry unique benefits for CL, but I’m afraid that the current version of this paper does not convincingly demonstrate this.
(A related issue is that the use of shape information makes the benchmarking experiments less controlled. The reason that methods that are compared against do not use shape information might well not be because “they did not think to do so”, but rather because they wanted to test specific approaches in isolation.)

Although the selected values for all hyperparameters are reported in Appendix D, it is not well described how these hyperparameters were selected for the different methods. It is stated that grid searches were performed, but it is not reported which hyperparameter values were considered for each method. Another drawback is that DUCA has relatively a lot of hyperparameters (which are set in a way that violates the CL principle of only seeing data in a certain sequence), more than most of the other methods that are compared with. Both of these are reasons contributing to why results of the benchmarking experiments by themselves are of limited value.

The paper presents DUCA as a “general” method for continual learning (among others by calling it a “framework for continual learning”). However, DUCA relies on the Sobel shape extraction algorithm, and as such its applicability seems to be limited to the image domain. Perhaps it could be argued that shape is merely used as an example of an inductive bias, and that for other modalities other inductive biases could be used. However, I do not think this is self-evident. I think either the authors should ensure it is clear that their proposed method is specific to the domain of images, or they should demonstrate that the underlying principle can also be successfully applied in other domains.

On page 6 the authors state: “As is evident from the different CL methods in the literature, the improvement in performance has been saturated on all variants of MNIST.” This is a very strong statement, but no references are provided for it. It also seems to me this statement is clearly wrong as I can think of many variations with MNIST for which performance has not saturated (e.g., when restricting memory, computation, amount of training samples). In fact, to strengthen this paper, I would encourage the authors to additionally include experiments on relatively simple datasets (for example MNIST), to provide results that are more easily reproducible and interpretable.

The way this paper describes class-, domain- and task-incremental learning seems rather simplistic and not in line with how these types of continual learning have been defined in Van de Ven et al. (2022, Nat Mach Intell). For example, on page 13 limitations of class-incremental learning are listed as “the assumption of the same number of classes across different tasks, the absence of reappearance of classes, and the sample distribution per class being well balanced”, but I don’t think these stated assumptions are part of the definition of class-incremental learning. Perhaps it could instead be argued that recent papers sometimes tend to make these assumptions when studying class-incremental learning?
I think the authors should also be careful with calling these assumptions “limitations”. For example, for DN4IL, the authors make simplifications relative to DomainNet; it could be argued these simplifications make DN4IL “less realistic” than DomainNet, but the authors will probably agree that these simplifications are not “limitations”.

Minor issues:
- “class incremental learning” should be “class-incremental learning” (same with task- and domain-incremental learning, see for example https://www.grammarly.com/blog/hyphen/)
- The first sentence of section 2.4 introduces “Cognitive Continual Learner (CCL)” but probably meant to say DUCA?
- The cited Gheiros et al paper was published at ICLR 2019, not ICLR 2018

---

> ### Author Response · Authors · 2023-09-06
> **Response to Reviewer jNRv -1/2**
>
> We extend our sincere gratitude to the reviewer for dedicating their valuable time and effort to provide insightful comments and feedback. We have provided the answers below and if you have any further questions or require additional information, please do not hesitate to contact us. Thank you.
>
> >  Split CIFAR could help towards this goal, but by themselves such experiments are of limited value, especially if there are concerns regarding how controlled they are.
>
> We appreciate the reviewer's feedback and concerns. While it is true that achieving state-of-the-art performance on benchmark datasets like variants of Split CIFAR may not be the ultimate goal of continual learning (CL) research, it serves as an important starting point. These benchmarks provide a standardized and controlled environment to evaluate and compare different CL methods. By demonstrating superior performance on these benchmarks (used in literature), we establish the effectiveness and competitiveness of the proposed approach, which is a crucial step towards building successful CL algorithms. Furthermore, we understand the importance of practical applicability and real-time implementation of CL methods. While we present benchmark results on Split CIFAR, we also introduce a challenging domain incremental learning dataset (DN4IL) to evaluate the proposed approach in more realistic scenarios. We have also included the MNIST results in the table here (and Table 5 in Appendix). By demonstrating robustness across different domains and addressing distribution shifts, we establish the efficacy and potential applicability of DUCA in practical CL settings. We welcome any specific suggestions or feedback from the reviewer for us to implement and present.
>
> > Contribution and novelty of shape
>
> The novelty of our proposed approach extends beyond the mere inclusion of shape information; it also resides in the intricate integration of the shape-based model within the architecture. Merely augmenting the primary training set with edge-detected or shape-emphasized images does not guarantee optimal results.
>
> To substantiate, we also provide more baselines in the table below (and also in Table 4 in Appendix).
> We add shape as another augmentation filter on other baselines (ER and DER++) and show the results .On SeqCIFAR100 the performance decreases highlighting the fact that merely adding both distributions may not always yield optimal results, underscoring the need for a more nuanced approach. On DN4IL, the results are better as the dataset is more complex and shape is a more discriminative feature. Indeed, shape information serves as a valuable complement to RGB data. The simple addition of shape, whether as supplementary data or through image augmentation, proves insufficient. The key lies in effectively guiding the network, which excels at learning visually rich information from RGB, to also acquire the capacity to discern and utilize semantic information. This approach is crucial for preventing shortcut learning and ensuring better generalization
>
> | Method | Specifics | Seq- CIFAR100 |  | DN4IL |  |
> |---|---|:---:|:---:|:---:|:---:|
> |  |  | 200 | 500 | 200 | 500 |
> | ER | Original | 21.40 | 28.02 | 26.59 | 31.01 |
> |  | RGB & Shape* | 19.47 | 23.96 | 27.45 | 33.44 |
> | DER++ | Original | 29.60 | 41.40 | 34.75 | 41.87 |
> |  | RGB & Shape* | 24.40 | 34.30 | 36.55 | 40.99 |
> | DUCA | Original | 45.38 | 54.27 | 44.23 | 49.32 |
> |  | -SM (RGB & Shape*) | 24.34| 32.64| 36.80 | 43.88 |
> |  | -SM -IBL (RGB & Shape*) | 19.47 | 23.96 | 27.45 | 33.44 |
> |  | -IBL (RGB & Shape) | 42.01 | 49.55 | 40.75 | 43.99 |
>
> We have conducted additional ablation experiments on DUCA to gain deeper insights. (1) DUCA without the inductive bias learner (-IBL), where we add shape as an augmentation instead of using another network. The results show that while shape is beneficial, the manner in which it is integrated into the training scheme plays a critical role. (2) we exclude the semantic memory module (-SM), retaining only the working and shape networks. These experiments help us discern the individual impact of each architectural component. The results underscore the significant contributions of both IBL and SM in enhancing performance. Interesting to note is also that on complex datasets (DN4IL), the impact of IBL is higher as the shape proves more beneficial. The learning of shape (and RGB both) in DUCA proves more optimal and is also not dataset dependent.
>
> >  grid searches for hyperparameters
>
> In response to your query, we have included the search grid utilized for hyperparameter tuning, which can be found in Table 6 within the Appendix for your reference.
> We agree that hyperparameter tuning can be a challenging and time-consuming task . However, we have designed DUCA to be relatively stable across different datasets and settings, so that it does not require extensive fine-tuning. We found that the trends of the parameters were similar and complemented each other.

---

> > ### Author Response · Authors · 2023-09-06
> > **Response to Reviewer jNRv -2/2**
> >
> > > DUCA limited to image domain
> >
> > We acknowledge the reviewer's concerns about the applicability of DUCA beyond the image domain. While our paper focuses on the image domain and utilizes the Sobel shape extraction algorithm as an example of an inductive bias, the main contribution of the paper lies in proposing a new learning mechanism based on the integration of inductive bias and memory systems. The DUCA framework can be extended to other domains by selecting appropriate inductive biases relevant to those domains. We also added in the paper (in Conclusion section) that while our current implementation and evaluation are specific to the image domain, the underlying principles and mechanisms of DUCA can be adapted to different modalities.
> >
> > > MNIST results
> >
> > Thank you for the suggestion. While it is true that there are variations of the MNIST dataset where performance can still be improved, the intention of the statement was to highlight the observation that significant progress has already been made in improving performance on standard variations of the MNIST dataset in the context of continual learning. We agree with the reviewer's suggestion to include experiments on datasets, such as MNIST, to provide more easily reproducible and interpretable results.
> > Hence, we also report the results of MNIST here and in Table 5 in Appendix. We utilize Rotated-MNIST to evaluate the domain-incremental setting. The results show that DUCA still fares better than other techniques.
> >
> > | Method | 200 | 500|
> > |---|---|---|
> > |ER | 85.01±1.90 | 88.91±1.44|
> > |DER++|90.43±1.87|92.77±1.05|
> > |CLS-ER| 92.26±0.18 | 94.06±0.07|
> > |DUCA| **93.53±0.21** | **94.98±0.04**|
> >
> > > Description of incremental learning settings
> >
> > We acknowledge the concern of the reviewer. Our motivation to test on GCIL-setting was influenced from “Mi, Fei, et al. "Generalized class incremental learning”, who use these assumptions as reasons to formulate the generalized-class incremental framework. We acknowledge that we can refrain from categorizing these assumptions as limitations of class-il, and instead employ GCIL as the benchmark for assessing a more encompassing learning setting. We have made changes in the paper.
> >
> > > Minor issues
> >
> > Thank you for the comments. We have fixed these issues in the text (all changes are in blue in the paper).

---

> > > ### Comment · Reviewer_jNRv · 2023-09-11
> > > **Response to author rebuttal**
> > >
> > > I thank the authors for their rebuttal; in particular for including MNIST results and for including additional comparisons showing that the way in which shape information is used is important. While both of these additions strengthen the paper, some concerns are not yet fully addressed.
> > >
> > > **Is the benefit of using shape information in the proposed way specific to CL?**
> > >
> > > The new experiments in Appendix C (Table 4) show that the way in which shape information is used is important, but these experiments do not address the question whether the observed benefit from using shape information in this way is specific to CL. That is, does using shape information in this way carry benefits that are unique to CL, or does it boost image classification performance in general? The authors briefly discuss this point in response to Reviewer 3ogS by referring to Figure 3 for some anecdotal evidence, but I could not find this discussed in the revised paper. I encourage the authors to also discuss this point in the paper itself.
> > >
> > > ***“These benchmarks [Split CIFAR] provide a standardized and controlled environment to evaluate and compare different CL methods.”***
> > >
> > > I agree that these benchmarks *could* provide this, but my concern is that in this paper not all comparisons are performed in a “standardized and controlled” way. For example, some methods use larger parameter budgets or more hyperparameters that are tuned in a problematic way (see below) than others. I encourage the authors to be more upfront about these differences between the compared methods in the paper.
> > >
> > > ***“We agree that hyperparameter tuning can be a challenging and time-consuming task.”***
> > >
> > > My concern is not that the tuning of CL hyperparameters is time-consuming, but rather that it is done in a way that violates the CL principle of only setting data in a certain sequence. For a discussion of the issue with selecting hyperparameters for CL with a grid search in the way done by the current paper, see for example Chaudhry et al., 2019 *ICLR* (section 2; https://arxiv.org/abs/1812.00420).
> > >
> > > ***“While our paper focuses on the image domain and utilizes the Sobel shape extraction algorithm as an example of an inductive bias, the main contribution of the paper lies in proposing a new learning mechanism based on the integration of inductive bias and memory systems.”***
> > >
> > > As also indicated in my original review, I do not think that it is self-evident whether or how “the principles and mechanisms of DUCA” can be adapted to other modalities. If the authors want to make a claim like this, or if they want to keep the (suggestive) remarks about their framework being general, they should demonstrate that the underlying principle of their method can indeed be successfully applied in other domains.
> > >
> > > Even if the authors only want to speculate that the principles and mechanisms of their proposed method could generalize to other inductive biases, I think it is still only appropriate to do so if the authors at least discuss some examples of other modalities and inductive biases they believe their approach could be applied to.
> > >
> > > **Description of incremental learning settings.**
> > >
> > > It is still not clear where the listed assumptions of (1) the same number of classes per tasks, (2) the absence of reappearance of classes and (3) the sample distribution per class being well balanced come from. As currently described in Appendix E, it is suggested these assumptions are part of the task-, domain- and class-incremental learning scenarios. However, I do not think that either the cited preprint (Van de Ven & Tolias, 2019 *ArXiv*) or its officially published version (Van de Ven et al., 2022 *Nat Mach Intell*; which would probably be the more appropriate source to cite) make these assumptions. The first two of these three assumptions are made in the “academic continual learning setting” described by Van de Ven et al. (2022), but not in the general case.
> > >
> > > Perhaps the authors could instead argue that recent papers sometimes tend to make these simplifications when studying class-incremental learning?
> > >
> > > Another comment is that on p14 the authors also still refer to the above simplifications as “limitations”.

---

> > > > ### Comment · Reviewer_jNRv · 2023-09-11
> > > > **Additional comment**
> > > >
> > > > I just noticed that the following claim is still included in the paper: "As is evident from the different CL methods in the literature, the
> > > > improvement in performance has been saturated on all variants of MNIST"
> > > >
> > > > However, it seems from their rebuttal that the authors agree that this statement is not true. Did the authors intend to remove or modify this statement?

---

> > > > > ### Author Response · Authors · 2023-09-11
> > > > > **Response to Reviewer jNRv - 3**
> > > > >
> > > > > Thank you for your prompt response.
> > > > > We inadvertently missed updating certain sections of the revised paper. We have now rectified this and submitted the updated version.
> > > > >
> > > > > -  Thank you pointing this out. We have added the text to highlight significance of the manner in which shape is integrated into the continual learning framework in Section 6.2, illustrated through Figure 3. We hope that these results, alongside the ablations in Table 4, emphasize the significance of shape within the context of continual learning. Should you require any further information, please let us know.
> > > > >
> > > > > - In regard to hyperparameters, we did not follow the evaluation setting outlined in [Chaudhry et al.'s 2019] paper. Instead, we adopted the hyperparameter tuning method utilized in all the baseline methods we compared against. We have added this detail in Section F.
> > > > >
> > > > > - Thank you for the feedback. We speculate that the principles and mechanisms of our proposed method could have broader applicability to various inductive biases. As an example, we have included potential inductive biases in the audio domain in Section H. Here, we suggest that leveraging spectrogram representations, which transform audio waveforms into visual data capturing both frequency and time-domain information, could be a viable option. Additionally, incorporating phonemes, which represent the fundamental sound units in spoken language and carry essential linguistic information, has the potential to enhance the framework's capabilities for tasks such as speech understanding, speaker identification, and language processing. Furthermore, we propose that the extraction of pitch and timbre features could be valuable for tasks like melody extraction, intonation analysis, and various music-related applications, providing valuable insights into the acoustic characteristics of audio signals.
> > > > >
> > > > > - We have made adjustments to the Class-IL assumptions and have also removed the reference to MNIST results being saturated.

---

### Review · Reviewer_n8Da · 2023-08-28

**Summary Of Contributions:**

This paper considers the continual learning problem in ANNs from the perspective of cognitive sciences. According to cognitive sciences, humans are able to perform a wide variety of difficult tasks due to the structure of the cognitive system, in particular multi-memory systems in the brain and incorporation of cognitive biases. The idea of the paper is to use these concepts in artificial neural networks to build a system that performs well in continual learning. The authors propose a Dual Cognitive Architecture, DUCA, which consists of three neural networks: the working model, the semantic memory, and the implicit bias learner that focuses on the shape of the image rather than the full RGB picture. The authors show that the proposed model works well on the split CIFAR dataset and the split MNIST dataset as compared to strong baselines from continual learning literature. The authors propose their own dataset for domain incremental learning and they also show that the method performs quite well there.  Finally, the authors perform a wide variety of ablation studies showing a more thorough understanding of when and why the model works in practice.

**Audience:**

Yes

**Claims And Evidence:**

No

**Requested Changes:**

Please add the shape-based baselines outlined in the first two points of the "Weaknesses" section.

**Strengths And Weaknesses:**

In general, I think the paper is quite interesting and should be valuable for the TMLR community.  However, before recommending acceptance, I would like to see some more baselines added just to understand the impact of the shape information on the final results.  If that issue is addressed, I would be happy to recommend acceptance.


Strengths
- The paper considers an important problem, from a perspective that I think has been fairly overlooked, i.e. cognitive sciences.
- The results shown in the paper are quite strong. The proposed method outperforms the baselines by a significant margin, both on the previously established benchmark as well as on the new benchmark.
- Additionally, the paper introduced the domain incremental learning dataset.  I think this is a valuable contribution that will be useful in further continual learning research.
- The paper is pretty clearly written, easy to understand, and well explained.


Weaknesses
- I think the paper doesn't show why exactly the proposed method works well. In particular, it might be that the module that uses the shapes as the input makes the classification problem easier in general. As such, I would like to ask the authors to introduce two new baselines. The first one is a network that simply learns only from the shape data.  So basically the bias model trained in isolation on the continual learning task. Another baseline would be a network that uses both the RGB data and the shape data as input, without the multi-memory structure introduced in the paper.
- Adding to the previous point, I think the author should try to understand how the shape information impacts the performance on any single task versus on the whole sequence. It might be that simply with the shape information we are able to better solve each task in separation and that's why we see the gains on the continual learning problem.
- The novelty of this work is not very high.  In particular, it seems like an incremental improvement over the CLS-ER method proposed previously by Arani et al. who use a similar multi-network approach.  In this work, the authors also use the inductive bias network, which seems somewhat incremental. On the other hand, for this venue, high novelty is not an important criterion for accepting the paper, so I don't think this is a good enough reason to reject this paper. However, I think this point should be noted.

---

> ### Author Response · Authors · 2023-09-06
> **Response to Reviewer n8Da**
>
> We value the considerate feedback provided by the reviewer. Please review the following response and advise if there are any additional questions or results we should incorporate.
>
> > Goal of DUCA
>
> The aim of DUCA was to explore an architecture inspired by the cognitive sciences in the continual learning framework. The innovative aspect of our proposed approach extends beyond just incorporating shape information; it also pertains to the manner in which the shape-based model is seamlessly integrated into the architecture. We aim for this cognitive-based architectural framework to serve as an inspiration for future research, encouraging the exploration of various inductive biases and the development of more efficient architectures.
>
> > Inclusion of new baselines and ablations
>
> Thank you for the suggestions, we provide more baselines and ablations for DUCA in the table below ( and also in Table4 in appendix). Specifically, we isolate and exclude the other elements of DUCA, such as IBL and SM. We train the base network (working model) under three distinct conditions: (1) only on shape images, (2) on RGB and shape images together, and (3) on RGB images with shape as an augmentation.
>
> | Method | Specifics | Seq- CIFAR100 |  | DN4IL |  |
> |---|---|:---:|:---:|:---:|:---:|
> |  |  | 200 | 500 | 200 | 500 |
> | DUCA | Original | 45.38 | 54.27 | 44.23 | 49.32 |
> |  | -SM -IBL (Shape only) | 18.33 | 21.98 | 27.89 | 31.57 |
> |  | -SM -IBL (RGB + Shape) | 20.57 | 25.20 | 31.52 | 35.68 |
> |  | -SM -IBL (RGB & Shape*) | 19.47 | 23.96 | 27.45 | 33.44 |
>
> The network trained only on shape “-SM -IBL (Shape only)” achieves lower results as shape information by itself is not enough. RGB images have rich information but sometimes fall into the pattern of shortcut learning, and an inductive bias such as shape provides global semantic information that complements it. Hence both the modalities are needed to achieve better generalization. And this complementary and useful knowledge can be leveraged by extracting and introducing it in the right technique.
>
> Secondly, training a single network on both distributions  “-SM -IBL (RGB + Shape)” or appending shape as an augmentation and training the base model “-SM -IBL (RGB & Shape*)”, also does not result in the optimal result. Interesting to note is that the results (when using shape) on  DN4IL, the results are better as the dataset is more complex and shape is a more discriminative feature. Thus just utilizing shape as augmentation varies on dataset and characteristics.
>
> The simple addition of shape, whether as supplementary data or through image augmentation, proves insufficient. The key lies in effectively guiding the network, which excels at learning visually rich information from RGB, to also acquire the capacity to discern and utilize semantic information. This approach is crucial for preventing shortcut learning and ensuring better generalization. In DUCA, training another network to learn shape and subtly supervising the working model with the information, performs better overall. The method is also not dependent on the dataset and provides a better balance to help improve generalization.
>
>
> > Performance on single task
>
> This is indeed a relevant point. For a more detailed explanation, please refer to Figure 3 in the paper, which shows the task wise performance for different methods. As seen, after the first task, DER++ has the highest performance  (62.3) and not DUCA. However, in continual learning settings the data arrives continuously and the focus is on both retention of old task performance while performing well on the current task. After the second task, the performance on first task decreases (44.5 -ER, 54.2-DER++, 57.2-CLSER and 62.9-DUCA), but with DUCA the forgetting is lesser, even though the performance on second task itself  is not the highest among four methods (61.4-DUCA). Thus, with DUCA, we observe a balance between stability and plasticity than other methods.

---

### Author Response · Authors · 2023-09-06
**General Response**

Dear reviewers,

We would like to express our gratitude for your time and valuable feedback. We've incorporated the suggested changes and uploaded the revised paper with the **modifications highlighted in blue for clarity.**

**Goal:**

The goal was to explore a architecture in continual learning inspired by the cognitive elements. The novelty of the proposed approach lies not only in the inclusion of inductive bias but also in how this knowledge is integrated into the architecture. As shown from various baselines and ablations, we show the efficacy of utilizing shape as an inductive bias and also the way it is integrated. We hope this cognitive-based architecture will inspire more works with different kinds of inductive biases and also architectures that are more efficient.

**Changes:**

- Table 4 and Section C in Appendix - additional baselines and ablation study to further substantiate the importance of the inductive bias and the different components in the DUCA architecture
- Table 5 and Section D - New comparison results on Rotated-MNIST data for domain-incremental setting
- Table 6 -  parameter search ranges that was used to perform grid-search to find the right hyperparameters.
- Table 9 - Number of parameters for each method

Apart from these results, we have made modifications in text based on the minor comments and feedback (in blue).

We would be happy to address any further questions or provide any additional information needed.

---

> ### Author Response · Authors · 2023-09-10
> **General Response - Note**
>
> A side note: We received an email from OpenReview indicating that a review had been edited. However, upon inspection, we couldn't identify any changes in the review history or the text itself. We would like to kindly request that if any changes were indeed made and remained unaddressed by us, please bring them to our attention. We appreciate your assistance. Thank you.

---

### Decision · Action_Editors · 2023-10-04

**Recommendation:** Accept with minor revision

**Comment:**

On balance, I believe that the authors have provided enough evidence to support their core claims and addressed most of the reviewer comments. However, I believe that there are still two important minor modifications that need to be made:

1) The abstract is not clear enough on what the model actually includes. Specifically, nowhere in the abstract is it noted that what the authors mean by "multiple memory systems" is a slow and fast learner for rehearsal (in-line with complementary learning systems theory), and that what they mean by "inductive biases" is specifically a shape bias. As such, a reader will not really understand the core model design from the abstract, even at a high level, so it is too vague. The authors should fix this before publication.

2) Though the authors inserted some language about moving this beyond the image domain, it is, in my judgement, insufficient. Specifically, the authors say, "...the underlying principles and mechanisms of DUCA can be adapted to different modalities. The DUCA framework can be extended to other domains by selecting appropriate inductive biases relevant to those domains." But, what would "appropriate inductive biases" be in other domains? Also, why would the other components of DUCA help in other domains? A few other sentences to suggest potential future directions to extend this work beyond the image domain would greatly help the conclusion of the paper.

**Audience:**

Yes, this paper would be interesting to TMLR readers interested in either continual learning or neuroscience-inspired approaches to ML.

**Claims And Evidence:**

This paper presents a model for continual learning that incorporates (1) multiple modules inspired by cognitive theories of a global workspace and two levels of representation (implicit and explicit), (2) an inductive bias for shapes, and (3) multiple memory systems specialised for fast and slow learning with rehearsal. The authors claim that the combination of these different brain-inspired mechanisms can help with continual learning. As evidence to support this claim, they present data showing improved performance on sequential CIFAR relative to other rehearsal-based models for continual learning in the literature. They also develop a new domain-incremental learning dataset, DN4IL, which is a subset of the DomainNet dataset designed to test performance in the face of distribution shifts, and show that their model again provides the best performance. They also use ablation studies to show that each of the three above design decisions appears to help with performance on these datasets.

Overall, the paper provides at least some reasonable evidence to support the claims. As noted by the reviewers, there would ideally be more evidence that (1) this strategy is not specific to vision tasks, (2) stronger evidence for the specific contribution made by each of the model design choices. But, there is at least some clear evidence for the core claims, and whether this evidence is convincing is at least partially a subjective question.

---

> ### Author Response · Authors · 2023-10-13
> **Camera Ready Version**
>
> Dear Action Editors and Reviewers,
>
> We extend our sincere gratitude for your time and effort in providing invaluable suggestions, constructive feedback, and encouraging remarks. We have implemented all the requested changes and have now submitted the revised PDF. Additionally, we have included links to the code and the new dataset.
>
> Regards
> Authors